# Serum FHR1 binding to necrotic-type cells activates monocytic inflammasome and marks necrotic sites in vasculopathies

Sarah Irmscher[1,7], Silke R. Brix[2,7], Svante L.H. Zipfel[3,7], Luke D. Halder[1], Sibel Mutlutürk[1], Sonia Wulf[4], Evaldas Girdauskas[3], Hermann Reichenspurner[3], Rolf A.K. Stahl[2], Berit Jungnickel[5], Thorsten Wiech[4], Peter F. Zipfel [1,6] & Christine Skerka [1]

Persistent inflammation is a hallmark of many human diseases, including anti-neutrophil cytoplasmic antibody-associated vasculitis (AAV) and atherosclerosis. Here, we describe a dominant trigger of inflammation: human serum factor H-related protein FHR1. In vitro, this protein selectively binds to necrotic cells via its N-terminus; in addition, it binds near necrotic glomerular sites of AAV patients and necrotic areas in atherosclerotic plaques. FHR1, but not factor H, FHR2 or FHR3 strongly induces inflammasome NLRP3 in blood-derived human monocytes, which subsequently secrete IL-1β, TNFα, IL-18 and IL-6. FHR1 triggers the phospholipase C-pathway via the G-protein coupled receptor EMR2 independent of complement. Moreover, FHR1 concentrations of AAV patients negatively correlate with glomerular filtration rates and associate with the levels of inflammation and progressive disease. These data highlight an unexpected role for FHR1 during sterile inflammation, may explain why FHR1-deficiency protects against certain diseases, and identifies potential targets for treatment of auto-inflammatory diseases.

[1] Department of Infection Biology, Leibniz Institute for Natural Product Research and Infection Biology, Adolf-Reichwein Str. 23, 07745 Jena, Germany. [2] III. Medical Clinic, University Medical Center Hamburg-Eppendorf, Martini Str. 52, 20246 Hamburg, Germany. [3] Clinic for Heart and Visceral Surgery, Center of Heart Diseases, University Hospital Hamburg-Eppendorf, Martini Str. 52, 20251 Hamburg, Germany. [4] Institute of Pathology, University Hospital Hamburg-Eppendorf, Martini Str. 52, 20251 Hamburg, Germany. [5] Department of Cell Biology, Institute of Biochemistry and Biophysics, Friedrich-Schiller University, Hans-Knöll Str. 2, 07745 Jena, Germany. [6] Faculty of Biological Sciences, Friedrich-Schiller University, Fürstengraben 1, 07743 Jena, Germany. [7]These authors contributed equally: Sarah Irmscher, Silke R. Brix, Svante L.H. Zipfel. Correspondence and requests for materials should be addressed to C.S. (email: christine. skerka@hki-jena.de)

Cellular stress is a spontaneous event triggered by injury or infection and is a characteristic of inflammatory diseases, such as anti-neutrophil cytoplasmic antibody-associated vasculitis (AAV)[1] and atherosclerosis (AS)[2, 3]. Necrotic cells act as unmodified damage-associated molecular patterns (DAMPs), which activate innate immunity and, together with membrane-anchored molecules, recruit and activate the inflammasome in immune cells, such as monocytes or neutrophils[4]. Pro-inflammatory cytokine IL-1β can be released via a cell death process (pyroptosis) which is a highly inflammatory event[5], or via hyperactivated cells where cells maintain viability[6]. In both cases pore forming gasdermin is involved[7]. Secretion of IL-1β, IL-6, IL-8, and TNFα recruits phagocytic leukocytes and activates stem cells to replace dead cells[8]. However, sustained immune responses can seriously damage host tissues and thus cause various dise

AAV is a systemic autoimmune disease characterized by autoantibodies specific for proteinase 3 (PR3) and myeloperoxidase (MPO) expressed by neutrophils. Histopathology commonly shows pauci-immune crescentic glomerulonephritis[1]. Activation of the alternative complement pathway amplifies the recruitment, priming, and activation of neutrophils, thereby creating a self-amplifying inflammatory loop that results in destructive and necrotizing vascular injury. In AAV, small- and medium-sized blood vessels, such as those in the kidney and lung, are infiltrated by immune cells and eventually destroyed[9].

AS is a chronic vascular disease and especially in its form of coronary artery disease the most common cause of death in developed countries. Lipid deposits form in regions of arteries with disturbed blood flow, leading to formation of atherosclerotic plaques comprising necrotic cells, which trigger sterile inflammation. Rupture of plaques can cause myocardial infarction or stroke. Patients with AS show high activation of the inflammasome 3 (NLRP3) in the aorta[2, 3].

Complement senses DAMPs and microbe-associated molecular patterns (MAMPs). Complement activation destroys microbes and induces phagocytosis and clearance mechanisms also of damaged and dead human cells. Factor H (FH), the main regulator of the alternative complement pathway, is crucial in the protection of self surfaces from the damaging effects of complement[10]. In contrast to these well studied vital functions of FH, the central function of the factor H related protein FHR1 remains unclear to date. FHR1 comprises five short consensus repeats (SCRs) but lacks the regulatory domains found in FH. In vitro assays revealed that FHR1 inhibits the terminal complement pathway in the absence of FH[11]. In addition, FHR1 competes with FH to regulate FH inhibitory activity[12, 13]. However, the composition and concentration of FHR1 suggest a more specific role in immunity. The N-terminal SCRs, named SCR1–2, within FHR1 are 36 and 45% identical to FH/SCR6–7, respectively, and contain a hybridization domain[13, 14]. The C-terminal SCR3–5 domains are 100/97, 100 and 98% identical to FH/SCR18–20, respectively, and bind C3b, C3d, and heparin[11]. FHR1 circulates in serum as homodimers and forms heterodimers with FHR2[14], which are contained in so-called FH-related proteins associated lipoprotein particles (FALPs)[15]. Genetic deletion of a chromosomal fragment comprising CFHR3-CFHR1 genes (ΔFHR1/3) confers protection against IgAN[16] and AMD[17], but susceptibility to systemic lupus erythematosus (SLE)[18] and atypical HUS[19]. The reason for these opposing associations between FHR1 and different diseases is still unclear, although likely ascribed to an as-yet-unknown function of FHR1. Here, we show FHR1 binding to necrotic-type cells and thereby inducing sterile inflammation, which is different to pyroptotic induced necroinflammation[20].

## Results

### FHR1 induces pro-inflammatory cytokine secretion.
In a previous study we demonstrated that FH binds to oxidized lipid deposits and inhibits complement activation and inflammatory responses[21]. To investigate whether FHR1 also modulates inflammation, we coated microtiter plates with FHR1 and incubated it with freshly isolated human peripheral blood monocytes in normal human serum (NHS) with or without lipopolysaccharide (LPS). Cytokine concentrations in the supernatant were measured after 20 h. The results showed that FHR1 alone strongly induced release of the pro-inflammatory cytokine IL-1β from monocytes, and increased LPS-triggered secretion of IL-1β (Fig. 1a). By contrast, immobilized FHR2, FHR3, FHL-1, FH, and BSA failed to induce IL-1β production (Fig. 1b, c). FHR1-induced IL-1β in a dose-dependent manner (0.6–5 µg ml[−1]) (Fig. 1d) as early as 3 h after the start of co-incubation (Fig. 1e). Inflammatory responses were triggered by the C-termini of FHR1/SCR3–5 and also FH/SCR19–20, as demonstrated in a similar assay. The N-terminus (SCR1-2) of FHR1 failed to induce IL-1β (Fig. 1f). Immobilized FHR1 did not trigger pyroptosis[22] as seen by no release of the enzyme lactate dehydrogenase (LDH) and full cell viability, measured via cell titer blue assay. (Fig. 1g). Similar to FHR1, immobilized mouse FHR1 homolog FHRB (Supplementary Fig. 1a) induced IL-1β secretion by mouse monocytes (Fig. 1h) in mouse serum. In parallel with IL-1β induction, FHR1-triggered secretion of pro-inflammatory cytokines IL-18, TNFα, and IL-6 (Fig. 1i–k), but not IL-8 (Supplementary Fig. 1b). It also inhibited secretion of the anti-inflammatory cytokine IL-10 by LPS-stimulated monocytes (Fig. 1l). Similar to IL-1β, TNFα was released by monocytes after 3 h of co-incubation with immobilized FHR1 (Fig. 1m).

Because FHR1 circulates in human serum, we next measured its pro-inflammatory function in a fluid phase. The microtiter plate surface was blocked and monocytes, FHR1, and NHS were added. In this setting, FHR1 had no effect on IL-1β (Fig. 2a) or IL-10 (Supplementary Fig. 1c) production. Thus, immobilization of FHR1 was necessary to trigger the pro-inflammatory response. Furthermore, pro-inflammatory function of FHR1 was lost without NHS, heat-inactivated or EDTA-containing NHS (Fig. 2b, and Supplementary Fig. 1d). FHR1-induced IL-1β release at low NHS concentrations (0.25%); which increased at higher NHS concentrations (Fig. 2c). In summary, surface-bound (not free) FHR1 induces secretion of IL-1β and inhibits IL-10 release by monocytes in the presence of active NHS.

### FHR1-induced inflammation is independent of complement.
Having shown that FHR1 requires active NHS to trigger secretion of IL-1β and to inhibit IL-10 production by monocytes, we asked whether the complement system is essential for its pro-inflammatory function. We found that FHR1 increased secretion of IL-1β and blocked IL-10 release in the presence of C3-depleted or C3- or C5-inhibited NHS (Fig. 3a, and Supplementary Fig. 2a) and with blocked CR3 or C3aR (Supplementary Fig. 2b). Neither C3 nor C5 was involved in FHR1 function. Blocking the classical- and the lectin-induced complement pathways (EGTA-treated or C1q-depleted NHS) or the alternative pathway (factor B- or P-depleted NHS) did not inhibit FHR1 activity (Fig. 3b, c, and Supplementary Fig. 2c, d). Thus, FHR1-mediated inflammation is independent of the complement pathways.

### FHR1 activates NLRP3 via the EMR2 receptor.
To find out how FHR1 induces the production and release of IL-1β[23], we first inhibited NF-κB or caspase-1 and found that FHR1-mediated release of IL-1β was completely blocked (Fig. 3d, e). FHR1-induced transcription of pro-IL-1β and TNFα exclusively in the

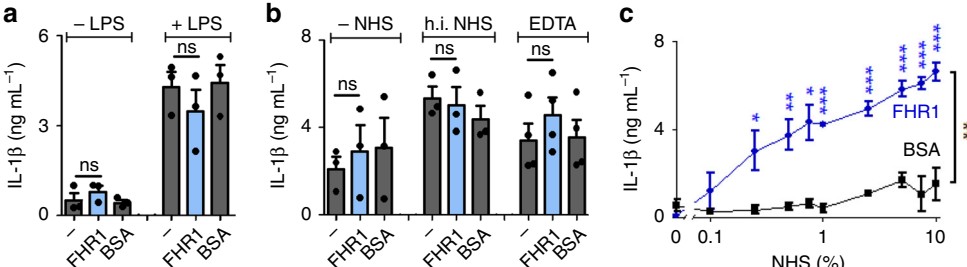

**Fig. 1** FHR1 induces inflammation. **a** Immobilized FHR1 induces IL-1β secretion by monocytes and increases IL-1β production by LPS-stimulated monocytes in the presence of NHS. **b**, **c** In contrast, FHR2, FHR3, FHL-1, FH, and BSA fail to induce IL-1β secretion. **d** Immobilized FHR1 reduces IL-10 secretion and increases IL-1β secretion by LPS-stimulated monocytes in a dose-dependent manner. **e** Immobilized FHR1 triggers IL-1β release from monocytes as early as 3 h after the start of incubation. **f** FHR1 SCR3-5 and FH SCR19-20, but not FHR1 SCR1-2, trigger IL-1β release by monocytes exposed to ΔFHR1/3 NHS. **g** Monocytes remain healthy after incubation with FHR1, as demonstrated by very low release of LDH (left) and full cell viability (right). Maximum LDH release (maxLDH) was measured after lysis of cells and lost viability with Nigericin sodium salt or Triton X-100 treatment via cell titer blue (CTB) assay. **h** FHRB induces IL-1β release by mouse monocytes exposed to mouse serum. Treatment with NLRP3 inhibitor (MCC950) inhibits FHRB-induced IL-1β release. **i** Immobilized FHR1 induces secretion of IL-18, **j** TNFα and **k** IL-6, and **l** reduces IL-10. **m** Immobilized FHR1 triggers release of TNFα from monocytes as early as 3 h after incubation. Data in **a–m** represent the mean ± SEM. of three to five independent experiments with different donor cells. *p ≤ 0.05, **p ≤ 0.01, ***p ≤ 0.001 (unpaired two-tailed t-test)

**Fig. 2** FHR1-induced inflammation depends on immobilization and NHS. **a** Unbound FHR1 fails to increase secretion of IL-1β by monocytes. **b** In the absence of NHS or in the presence of heat-inactivated (h.i.) and EDTA-inactivated NHS (EDTA), immobilized FHR1 does not induce IL-1β secretion. **c** FHR1 induces secretion of IL-1β upon exposure to 0.25% NHS. Increasing the concentration of NHS increases IL-1β release. Data in **a–c** represent the mean ± SEM. of three to four independent experiments with different donor cells. *p ≤ 0.05, **p ≤ 0.01, ***p ≤ 0.001 (unpaired two-tailed t-test)

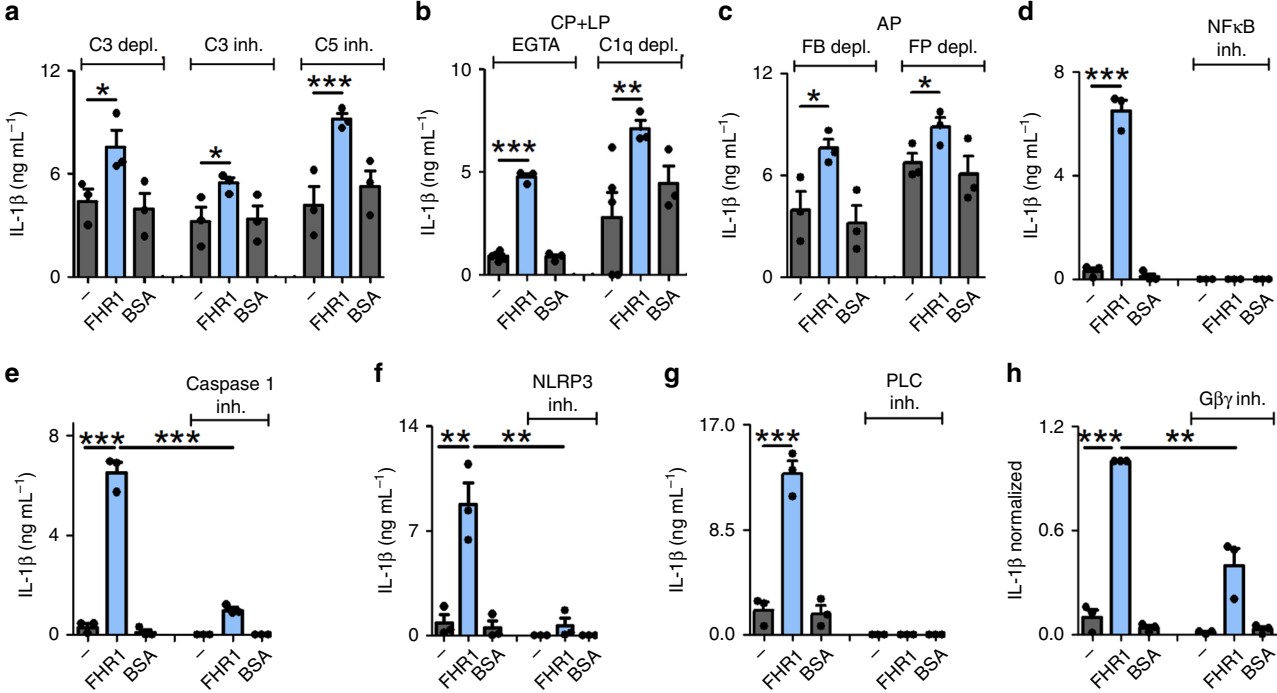

**Fig. 3** FHR1 activates NLRP3 in monocytes via the PLC pathway. **a** Immobilized FHR1 increases secretion of IL-1β in LPS-treated monocytes exposed to C3-depleted, C3-inhibited (Compstatin), or C5-inhibited (Eculizumab) NHS. **b** FHR1 stimulates secretion of IL-1β by EGTA-treated or C1q-depleted NHS, which blocks the classical (CP) and lectin pathways (LP) activation, respectively, as well as in **c** factor B (FB)- or factor P (FP)-depleted NHS (in which the alternative pathway (AP) is inhibited). **d** IL-1β induction by FHR1 is blocked when NF-κB is inhibited by BAY 11-7085. **e** IL-1β secretion is blocked by a caspase-1 inhibitor (VX-765), **f** by a NLRP3 inhibitor (MCC950), or **g** by a PLC inhibitor (U73122). **h** Blocking Gβγ subunit of GPCRs with Gallein reduces IL-1β induction by FHR1 by about 65% after 4 h incubation. Data in **a-h** represent the mean ± SEM. of three independent experiments with cells from different donors. *$p \leq 0.05$, **$p \leq 0.01$, ***$p \leq 0.001$ (unpaired two-tailed $t$-test)

presence of NHS, as shown by increased RNA transcription in monocytes (Supplementary Fig. 2e, f). However, inhibiting TLR2, TLR4, TLR6, CD14, CD36, RAGE, Dectin 1, or Fc-receptors did not interfere with FHR1-induced IL-1β secretion (Supplementary Fig. 2g–j and Supplementary Table 1). Thus, FHR1 in presence of active NHS induces the inflammasome via a specific receptor-mediated signaling pathway.

Caspase-1 is activated by NLRP1, NLRP3, NLRC4, AIM2 or Pyrin[24]. Incubating monocytes with a specific NLRP3 inhibitor completely inhibited FHR1- and FHRB-mediated IL-1β release, thereby confirming involvement of NLRP3 in the activation process (Fig. 3f, and Fig. 1h). To identify the FHR1-triggered pathway, we measured reactive oxygen species (ROS) as ROS release or cathepsin B trigger formation of the NLRP3 inflammasome. FHR1 failed to induce ROS release by monocytes (in contrast to the microbe *C. albicans*) (Supplementary Fig. 2k). Similarly, inhibiting K+ efflux[25] and cathepsin B did not affect FHR1-induced IL-1β production (Supplementary Fig. 2l). However, IL-1β release is blocked by phospholipase C (PLC) inhibitors (Fig. 3g, and Supplementary Fig. 2m) and substantially reduced by inhibition of Gβγ, the subunit of G protein-coupled receptor (GPCR) (Fig. 3h).

As many pro-inflammatory receptors induce PLC and Ca2+, we examined the FHR1 signaling pathways by RNA sequencing in monocytes derived from four different donors upon incubation with FHR1 and NHS for 4 h. We identified 522 monocytic genes upregulated in response to FHR1 (gene ontology enrichment analysis: 'immune response', 'cellular response to TNF', 'neutrophil chemotaxis', 'cellular response to IL-1', and 'signal transduction and inflammatory response' (Fig. 4a, b). 35 upregulated genes (6.7%) were inflammation related, including NF-kB, NLRP3, and calcium-signaling pathways (KEGG) (Fig. 4c,

d). Gene associated disease (GAD) analysis confirmed that patterns of upregulated genes by FHR1 are found in chronic renal failure, hypertension, myocardial infarction, atherosclerosis, stroke, and blood pressure (Fig. 4e). Furthermore, FHR1 upregulated the GPCR EMR2/ADGRE2, which was previously described to induce an inflammatory response via PLC and intracellular Ca2+ release[26]. Indeed, inhibition of EMR2 receptor completely blocked FHR1-induced IL-1β release (Fig. 4f). Subsequent interaction assays confirmed binding between recombinant EMR2 and FHR1 by ELISA and surface plasmon resonance technique (Fig. 4g, h). In summary, FHR1 activates PLC via EMR2, which stimulates Ca2+ release from the endoplasmic reticulum and activates NLRP3, followed by activation of caspase-1, which then cleaves pro-IL-1β to IL-1β[27] (Fig. 4i).

**FHR1 binds to necrotic cells and induces inflammation**. Having shown that surface-coated FHR1 induces the NLRP3 inflammasome, we asked when FHR1 mediates this function. Healthy or necrotic human endothelial cells (HUVECs) were incubated with FHR1 or purified FH and binding was analyzed by flow cytometry. FHR1, but not FH, bound to necrotic HUVECs and neither molecule bound to healthy cells (Fig. 5a, and Supplementary Fig 3a). Also, FHR1 did not bind to living or early apoptotic (induced by staurosporine) cells, but to necrotic (staurosporine), necroptotic (induced by Z-VAD-FMK, TNFα and Cyclohex-imide) and pyroptotic cells (induced by Nigericin sodium salt) (Supplementary Fig. 3b–d). Similar to FHR1, FHRB bound to necrotic mouse alveolar macrophages, but not to healthy cells (Fig. 5b, and Supplementary Fig. 3e). Thereby FHR1 bound to distinct spots on necrotic cells, which had lost cell surface integrity as seen by reduced wheat germ agglutinin (WGA) staining (Fig. 5c). When necrotic HUVECs were incubated in

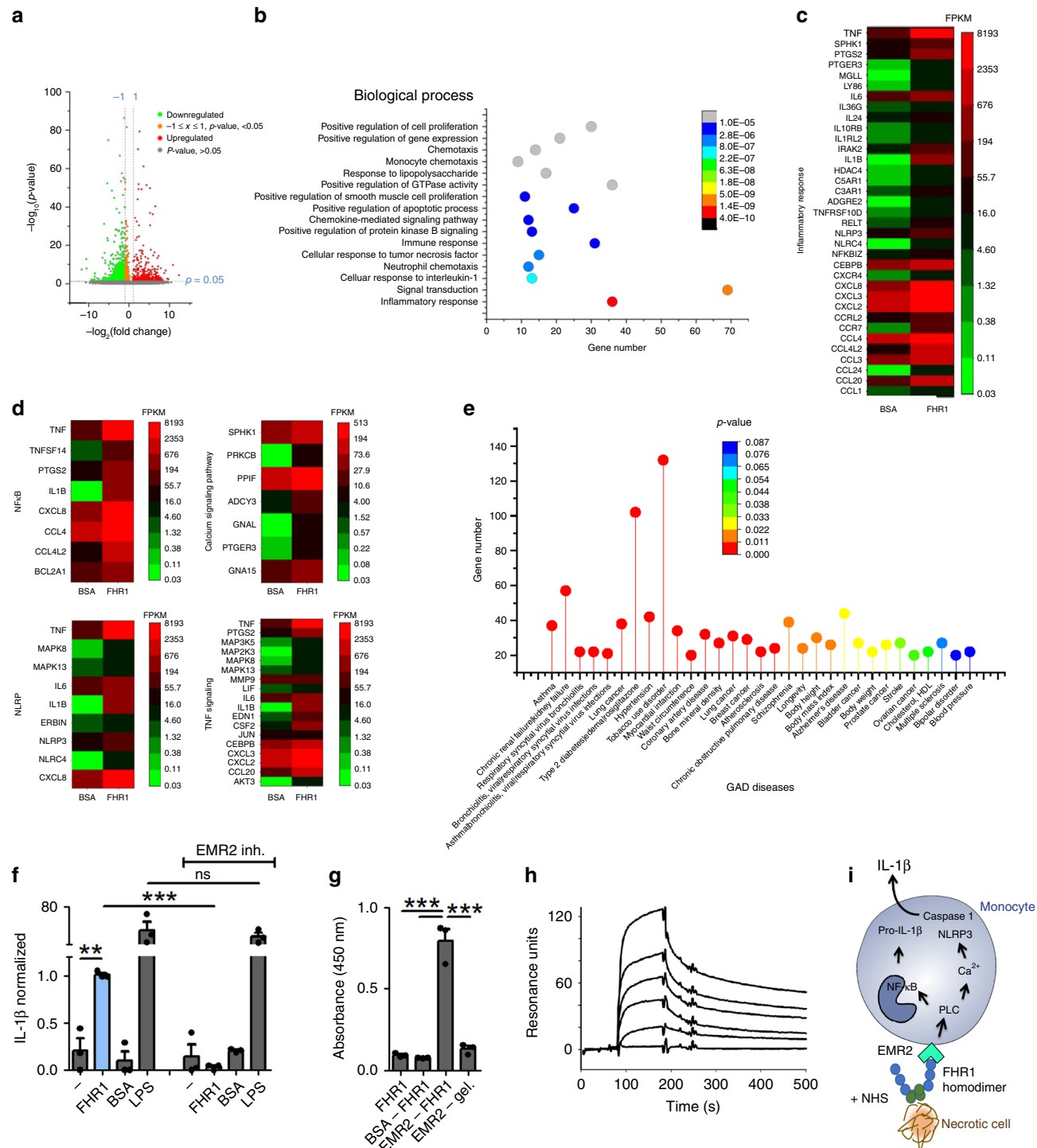

**Fig. 4** FHR1 binds to G protein-coupled receptor EMR2. **a** Genes differentially expressed in monocytes exposed for 4 h to FHR1 and NHS. **b** Biological processes involved in FHR1 induction (GO classification). **c** Inflammatory genes and **d** signaling pathways induced in monocytes by FHR1. **e** Upregulated genes in response to FHR1 show significant overlap with upregulated genes in certain diseases. **f** IL-1β induction by FHR1 is blocked by inhibition of EMR2 (αEMR2). **g** Recombinant EMR2 binds to FHR1 but not to gelatine (gel.) in ELISA. **h** The bond between EMR2 and FHR1 is in a dose-dependent manner shown via surface plasmon resonance technique. **i** Schematic overview of FHR1-induced IL-1β generation pathway in monocytes. Data in **a**–**h** represent the mean ± SEM of three to four independent experiments with cells from different donors. **p ≤ 0.01, ***p ≤ 0.001 (unpaired two-tailed t-test)

NHS, both FH and FHR1 from NHS bound to necrotic but not to living cells (Fig. 5d). In this case, binding of FH was explained by its interaction with C3b on the surface; by contrast, FHR1 bound to cells in the presence/absence of C3b. When bound to C3b (Supplementary Fig. 3f) via SCR3–5[11], FHR1 lost the ability to induce release of IL-1β (Fig. 5e). However, FHR1 alone attached

to necrotic cells via the N-terminal SCR1–2 domains, as shown by flow cytometry (Fig. 5f, and Supplementary Fig. 3g). Therefore we concluded that FHR1 domains SCR3–5 trigger inflammation when FHR1 is bound to necrotic cells.

To examine whether necrotic cell surface-bound FHR1 also induces inflammation, we incubated necrotic HUVECs with

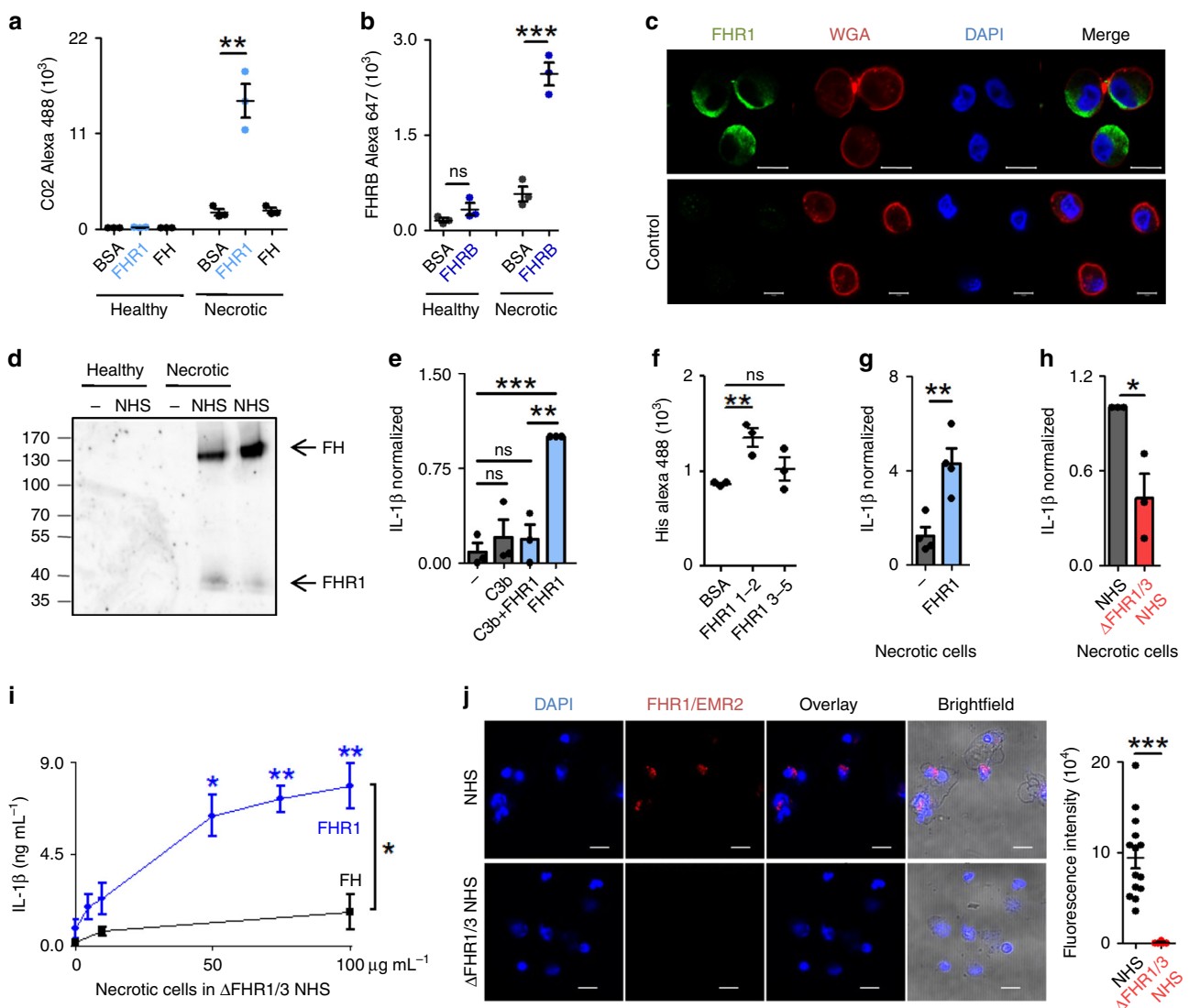

**Fig. 5** FHR1 binds to necrotic cells and triggers IL-1β release by monocytes. **a**, FHR1, but not FH binds necrotic HUVECs, as shown by flow cytometry. **b** FHRB binds to necrotic but not healthy mouse alveolar macrophages, FHRB antiserum was used for flow cytometry. **c** FHR1 (green) binds to necrotic cells. DNA (blue) is stained with DAPI and membrane (red) with WGA. Scale = 10 μm. **d** FHR1 and FH from NHS bind to necrotic, but not to healthy cells. One representative western blot of three is shown. **e** FHR1 bound to C3b does not induce IL-1β release by monocytes. **f** FHR1 SCR1–2 binds to necrotic HUVECs. **g** FHR1 bound to necrotic HUVECs triggers IL 1β in monocytes. **h** IL-1β release in response to necrotic HUVECs is reduced in ΔFHR1/3 NHS compared to NHS. **i** Reconstitution of FHR1 in ΔFHR1/3 NHS increased IL-1β release by monocytes. FH has no effect. **j** Complexes (red) of FHR1 (on necrotic cells) with EMR2 receptor (on monocytes) in whole blood. Total picture fluorescence intensity of single cells (n = 14 for FHR1/3, n = 10 for ΔFHR1/3) were analyzed using unpaired two-tailed t-test. Data in **a**–**i** represent the mean ± SEM of three to four independent experiments with different cell donors. *$p \leq 0.05$, **$p \leq 0.01$, ***$p \leq 0.001$ (unpaired two-tailed t-test)

FHR1 and incubated them with monocytes in presence of NHS. FHR1 bound to necrotic cells triggered a >four-fold increase of IL-1β release by monocytes when compared with necrotic cells alone (Fig. 5g). As expected, incubating necrotic cells with monocytes in NHS containing FHR1-triggered IL-1β release. The IL-1β level dropped by about 60% when monocytes were incubated in FHR1 deficient (ΔFHR1/3) NHS (Fig. 5h). Reconstitution of ΔFHR1/3 serum with FHR1 resulted in a strong release of IL-1β; this was not the case for FH (Fig. 5i). To confirm FHR1 interaction with EMR2, FHR1/EMR2 complexes were determined by proximity ligation assay (PLA) in an ex vivo whole blood model system with necrotic cells. FHR1/EMR2 complexes were identified on monocytes that were in contact with necrotic cells or cell debris when incubated in

FHR1 sufficient but not in ΔFHR1/3 blood (Fig. 5j). In contrast, FHR1 did not form complexes with TGFβRI (Supplementary Fig 3h).

**FHR1 binds to MDA-LDL and increases inflammation.** Malondialdehyd-modified low-density lipoproteins (MDA-LDL) are a marker of oxidative stress and are produced by damaged and necrotic cells[28]. FHR1 binds to MDA-LDL, but not to LDL or BSA (Fig. 6a) in a dose-dependent manner (Fig. 6b,c). Binding of FHR1 to MDA-epitopes expressed by necrotic HUVECs was confirmed using PLA (Fig. 6d). Similar to necrotic cells, FHR1 also bound to MDA-LDL via SCR1–2 (Fig. 6e). MDA-LDL-bound FHR1 increased secretion of IL-1β

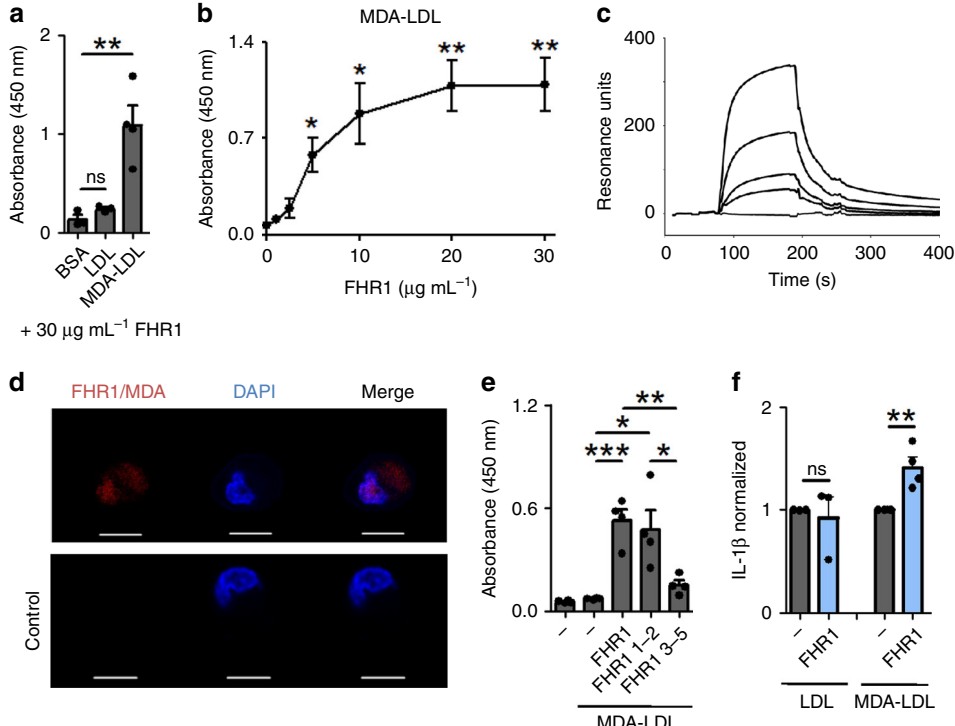

**Fig. 6** FHR1 binds to MDA-LDL on necrotic cells. **a** FHR1 binds to MDA-LDL, but not to LDL or BSA. **b** FHR1 binds to MDA-LDL in a dose-dependent manner as shown by ELISA and **c** surface plasmon resonance. **d** FHR1 binds to MDA epitopes on necrotic cells, as demonstrated by proximity ligation assay. Red dots denote FHR1-MDA complexes. DNA is stained with DAPI (blue). Control = one antibody. **e** FHR1 binds with its N-terminus FHR1 SCR1-2 to MDA-LDL. **f** FHR1 bound to MDA-LDL, but not LDL, increases IL-1β release by monocytes. Data in **a**–**f** represent the mean ± SEM. of three to four independent experiments with different cell donors. *$p \leq 0.05$, **$p \leq 0.01$, ***$p \leq 0.001$ (unpaired two-tailed *t*-test)

by monocytes by about 40% (Fig. 6f). Thus, FHR1 is a sensor of necrotic cell surfaces and induces inflammation.

**FHR1 binds to necrotic cells in vivo.** We next asked whether FHR1 is targeted to specific necrotic cells in vivo. Necrosis and inflammation are hallmarks of AAV and AS[29, 30]. Therefore, attachment of FHR1 to glomeruli in AAV patients and to atherosclerotic plaques in AS was examined by immunohistochemistry (IHC). Kidney tissue sections from patients diagnosed with AAV and glomerulonephritis were stained for FHR1. Granular positivity is seen for FHR1 in parts of the glomerular matrix that are in proximity to fibrinoid necrosis[31] but not on healthy tissue (Fig. 7a). Weak signals were detected for C3c and FHR2 in these areas (Supplementary Fig. 4a). In AS plaques on surfaces are formed by lipid accumulation and oxidation and contain necrotic cells. Staining of human plaques in the human artery and heart valve revealed specific binding of FHR1 to the plaques, but not to healthy cells close to the plaques (Fig. 7b). In contrast to FHR1, FHR2 showed low binding to the heart valve and is found predominantly in the intima of the artery (Fig. 7b).

Furthermore, FHR1/EMR2 but no FHR1/TGFβRI complexes were identified in tissue section from FHR1 sufficient AAV and AS patients but not in sections from ΔFHR1/3 AAV patients (Fig. 7c, and Supplementary Fig. 4b and 4c). In summary, FHR1/EMR2 complexes are formed in AAV and AS tissues.

**ΔFHR1/3 NHS from AAV patients show low IL-1β and CRP.** FHR1 binds to necrotic cells and induces inflammation. To determine the relevance of this process, we measured IL-1β concentrations in AAV patients. In a cohort of 314 AAV patients, 11 had ΔFHR1/3 deficiency, which is consistent with the

frequency in the normal healthy Caucasian population[32] (Supplementary Fig. 4d). AAV patients harboring FHR1/3 or ΔFHR1/3 are of similar age and gender (Supplementary Fig. 4e, f). When we compared IL-1β serum concentrations according to the presence/absence of FHR1, we found very low amounts of IL-1β in AAV patients with ΔFHR1/3 (0.6 ± 0.1 pg ml⁻¹), similar to those in healthy controls (0.4 ± 0.1 pg ml⁻¹). By contrast, in patients with active FHR1 significantly higher IL-1β concentrations were detected (1.4 ± 0.3 pg ml⁻¹, $p < 0.05$, unpaired *t*-test with Welch's correction) (Fig. 7d). IL-1β serum concentrations did not directly correlate with ANCA antibodies, in contrast to C-reactive protein (CRP) (Supplementary Fig. 4g, h). Therefore, CRP concentrations, which can substantially increase in response to IL-6, were analyzed in serum samples of AAV patients and revealed significant lower amounts in patient samples with ΔFHR1/3 (5.4 ± 0.4 mg dL⁻¹) in comparison to patients harboring FHR1/3 (36.9 ± 3.5 mg dL⁻¹, $p < 0.001$, unpaired *t*-test with Welch's correction) (Fig. 7e). In addition, we measured exact FHR1 concentrations in the serum samples by ELISA. AAV patients showed significantly increased FHR1 concentrations (69.8 ± 3.7 µg mL⁻¹) compared to healthy individuals (26.5 ± 2.3 µg mL⁻¹, $p < 0.0001$, unpaired *t*-test with Welch's correction) (Fig. 7f). As the previously published FHR1 concentration in healthy donors was 70–100 µg mL⁻¹ [11], we tested different protein sources for standard curves in the used ELISA kit and indeed, the FHR1 concentration in NHS varied by a factor of 1–3.5 (25–95 µg mL⁻¹). Also the IL-1β concentration increased with FHR1 serum concentrations ($p = 0.0598$, Spearman correlation) in AAV patients without immunosuppressive therapy (Supplementary Fig. 4i). Furthermore, the estimated glomerular filtration rate (eGFR) negatively correlated with increasing FHR1 concentrations (p < 0.005, Spearman

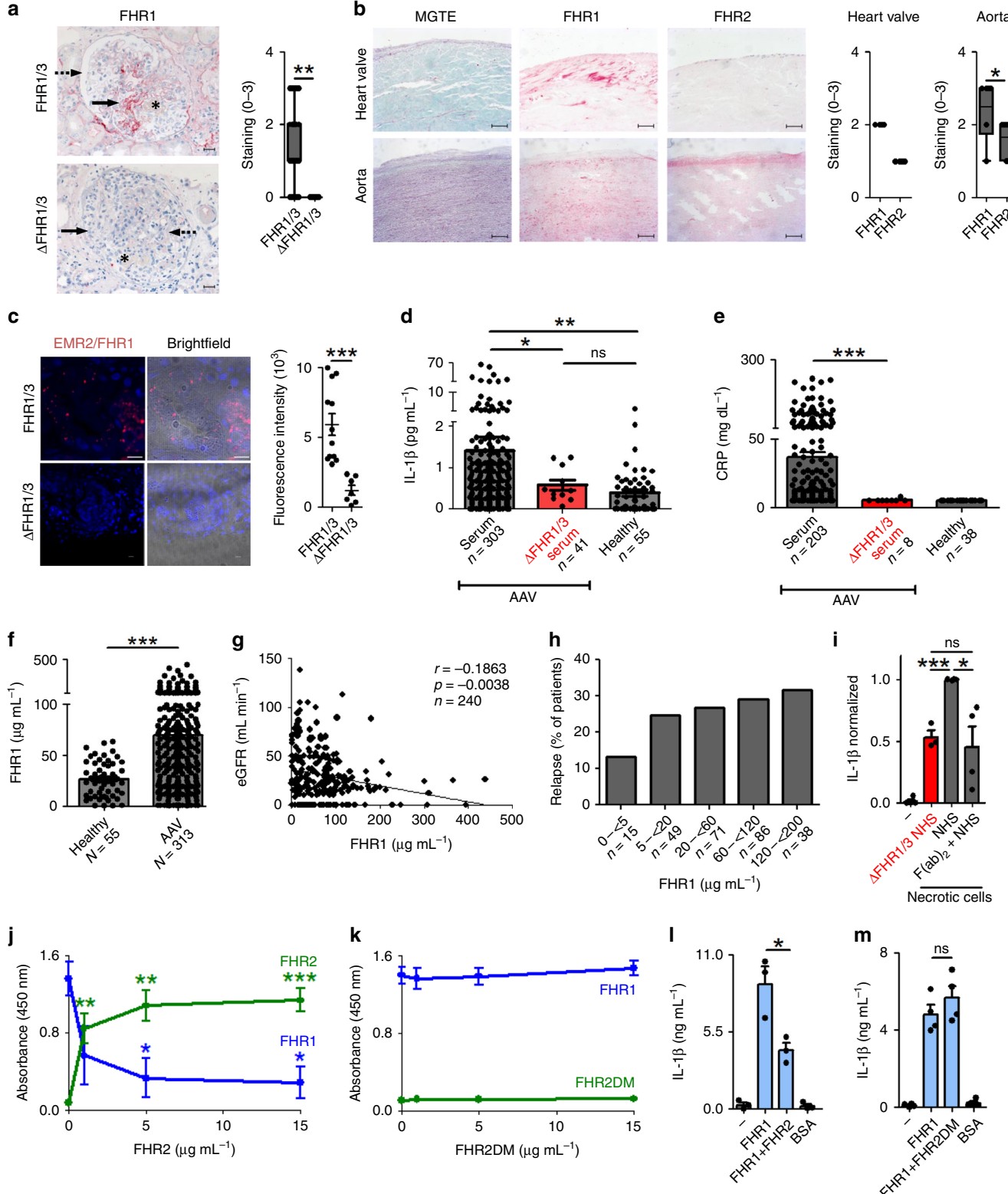

correlation) (Fig. 7g) and accordingly relapse (Fig. 7h) and death rates (Supplementary Fig. 4j) increased with growing FHR1 concentrations. PR3/MPO antibody positive patients with ΔFHR1/3 showed reduced fresh fibrinoid necrosis and cellular crescents compared to FHR1/3 harboring patients (Supplementary Fig. 4k, l). Altogether the data demonstrate an in vivo influence of FHR1 on inflammation and progression of the disease.

**Inhibition of FHR1**. As inflammation is reduced by 50% in ΔFHR1/3 NHS we asked whether FHR1 antibodies would decrease the inflammatory response in NHS. Monoclonal FHR1 F(ab)$_2$ fragments applied to NHS reduced the IL-1β release of monocytes by about 50%, thus to the same level as in ΔFHR1/3 NHS (Fig. 7i). In conclusion, inhibition of the FHR1 N-terminus by a monoclonal antibody reduced FHR1-mediated inflammation and provides a therapeutic option.

**Fig. 7** In vivo functions of FHR1. **a** In AAV, FHR1 attaches to necrotic cells (arrow), surrounding fresh fibrinoid necrotic lesions in the glomerulus (asterisk) but not in healthy tissue (discontinued arrow) (n = 79). Glomeruli derived from ΔFHR1/3 AAV patients (n = 9) lack FHR1 signals. Bars = 20 μm. **b** FHR1 (red) signal is found on macrophages and damaged smooth muscle cells in atherosclerotic plaques in human heart valve (n = 4, bars = 50 μm) and arteries (n = 6, bars = 200 μm) of AS patients. FHR2 staining is located to the intima of arteries. MGTE = Masson Goldner Trichrome Elastica. **c** EMR2/FHR1 complexes in kidney biopsies of FHR1/3, but not of ΔFHR1/3 AAV patients as shown by PLA assay. DNA is stained with DAPI. Twelve pictures of 3 FHR1/3 and seven pictures of 2 ΔFHR1/3 were analyzed (unpaired two-tailed t-test). Bar = 10 μm. **d** Serum IL 1β and **e** CRP with FHR1/3 compared to ΔFHR1/3 AAV patients (**d, e** unpaired two-tailed t-test with Welch's correction and Kruskal–Wallis Test). **f** FHR1 concentrations in healthy individuals and AAV patients (unpaired two-tailed t-test, Welch's correction). **g** Estimated glomerular filtration rate (eGFR) decreases (Spearman correlation) and **h** percentage of relapses of AAV patients increases with growing FHR1 concentrations. **i** Addition of F(ab)₂ fragments of a FHR1 antibody to NHS reduces monocytic IL-1β to a level comparable to inflammation in ΔFHR1/3 NHS triggered by necrotic cells. **j** Natural FHR1 monomers bound to FHR1 SCR1–2 and are replaced by FHR2, but **k** not by dimerization mutant FHR2DM. **l** FHR1 on necrotic cells in ΔFHR1/3 NHS increases IL-1 β, which is reduced by FHR2 but not **m** by FHR2DM. Data in **a–b** show box and whiskers from min to max with mean ± SEM. (Wilcoxon Signed Rank test) and data in i-m represent the mean ± SEM. of three to four independent experiments with different cell donors (unpaired two-tailed t-test). **a–m** *p ≤ 0.05, **p ≤ 0.01, ***p ≤ 0.001

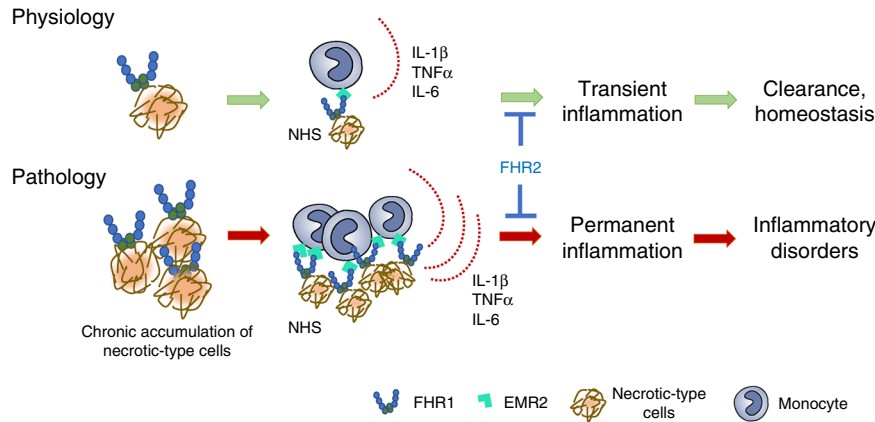

**Fig. 8** Overview. FHR1 binds to necrotic-type cells and induces inflammation in monocytes in NHS. Accumulation of necrotic-type cells recruits FHR1, which over-activates monocytes and induces inflammatory disorders

FHR1 forms homodimers and heterodimers with FHR2[13]. When SCR1–2 of FHR1 was immobilized and incubated with NHS, we found that serum-derived FHR1 bound to FHR1 SCR1–2. Addition of increasing amounts of FHR2 out-competed serum-bound FHR1 (Fig. 7j). By contrast, replacement of FHR1 was not observed when the dimerization motif of FHR2 was mutated (FHR2DM) (Fig. 7k, and Supplementary Fig. 4m). Addition of FHR2, but not FHR2DM, to the serum reduced the concentration of monocytes derived IL-1β by about 50% (Fig. 7l, m). These results demonstrate that FHR2 inhibits FHR1 activity by forming heterodimers, and that FHR2 may restrict FHR1-mediated inflammation.

## Discussion

Inflammation is a fundamental cellular response to harmful stimuli such as pathogens, trauma, and necrosis. In these cases, transient inflammation leads to clearance and healing, thereby maintaining cellular homeostasis. However, sustained immune responses can cause auto-inflammation, in which innate immune responses play the primary pathophysiological role[33]. Here, we identified a human inducer of inflammation, FHR1, which binds to oxidized LDL on necrotic-type human cell surfaces and strongly induces the NLRP3 inflammasome in monocytes via GPCR EMR2 exposed to NHS (Fig. 8). FHR1 deficiency, which is present in 4–7% of Caucasians[32], leads to a substantial reduction in IL-1β secretion (about 50%) in response to necrosis. Accordingly, AAV patients harboring ΔFHR1/3 show low IL-1β as well as CRP serum concentrations and are better regarding glomerular necrosis and relapses compared to patients with FHR1. This is confirmed by reduced necrotic inflammation (about 50%) upon addition of monoclonal FHR1 F(ab)₂ fragments to NHS in vitro.

These data demonstrate a dominant function of FHR1 in necrotic-type surface triggered inflammation, which may explain why ΔFHR1/3 protects against auto-inflammatory diseases such as IgAN[16] and AMD[17]. Both diseases are characterized by cell stress and pathological inflammatory processes with involvement of the NLRP3 inflammasome[34].

Immobilized FHR1 (as well as the murine homolog FHRB), but not FHR2, FHR3, FH, or the spliced variant of FH, FHL-1, induces the NLRP3 inflammasome in monocytes, which subsequently secrete pro-inflammatory cytokines. IL-1β and TNFα are central cytokines in immune cell communication and activation and as such recruit innate immune cells to the site of infection and cell damage. IL-6 triggers CRP release and IL-18 is responsible for generating interferon-γ (IFN-γ) and for increasing the cytolytic activity of natural killer cells and T-cells. To induce IL-1β, FHR1 binds to oxidized lipids on necrotic cells via its N-terminal domain, linking the pro-inflammatory features of oxidized lipids[35, 36] to NLRP3 activation in monocytes. Thereby FHR1 does not induce cell death such as pyroptosis, necroptosis or necrosis in monocytes and cytokines are secreted via intact lipid bilayers as previously described for hyperactivated macrophages[7]. The benefit of cell survival is that IL-1β together with secreted factors can act immunomodulatory. Upon binding, FHR1 likely changes its conformation such that the C-terminus becomes exposed and mediates the inflammatory response. As both the C-terminal domains of FHR1 (SCR3–5) and FH (SCR19–20) induce IL-1β secretion by monocytes, this suggests that the C-terminal domains of both proteins are hidden or protected when they are circulating in human plasma. Indeed, FHR1 is present primarily in so-called FALPs[15], which may shield the inflammatory activity of FHR1. Similarly, FH with a circular

structure was previously described, in which the C-terminus is folded back[37]. Like FH, FHR1 can bind to the TED domain in C3b via the C-terminal SCR domains[38]. When bound to C3b, FHR1 lacks pro-inflammatory activity, confirming the C-terminus as the main inflammatory actor in FHR1. In summary, FHR1 can activate the NLRP3 inflammasome in immune cells by binding directly via its N-terminus to oxidized lipids on necrotic cells.

FHR1 activates the NLRP3 inflammasome in monocytes via the PLC and $Ca^{2+}$ activation pathways. RNAseq data derived from FHR1-induced monocytes suggested involvement of GPCR EMR2/ADGRE2, which induces inflammatory responses in macrophages[26, 39]. Specific inhibition of EMR2 confirmed FHR1 action via this receptor on monocytes. Activation of NLRP3 via binding of FHR1 to EMR2 is dependent on an additional so far unknown serum signal. However, EMR2 has been shown to be cleaved by a self-catalyzed process characteristic of an autoproteolytic reaction[40] which might be triggered by the serum signal. EMR2 involvement in FHR1 function is supported by the fact that EMR2 is highly expressed in foamy macrophages in atherosclerosis[41, 42] and to correlate with disease severity and mortality in patients with cirrhotic liver disease[39].

Necrosis plays an important role in AAV and treatment of inflammation through aggressive immunosuppression is the established therapy[43, 44]. Also, in AS recent trials follow inhibition of inflammation[45]. ΔFHR1/3 was detected in 3.5% of AAV cases and IL-1β concentrations were not increased in these ΔFHR1/3 AAV patients; indeed, the concentrations were low and comparable to those measured in healthy controls. Moreover, the acute phase protein and inflammation marker CRP, which increases in response to IL-6, was significantly lower ($p < 0.001$, unpaired t-test with Welch's correction) in ΔFHR1/3 AAV samples compared to patients harboring FHR1. Noteworthy patients with ΔFHR1/3 had less fibrinoid necrosis and less cellular crescends compared to FHR1/3 harboring patients. In AAV patients the overall FHR1 concentration was much higher compared to healthy individuals. The FHR1 serum concentration associated with the progression of the disease as they negatively correlated with the glomerular filtration rates and with increased relapses and death rates. Also, FHR1 concentrations in AAV patients without immunosuppressive therapy showed increased IL-1β concentrations. These results confirm the pro-inflammatory function of FHR1 in AAV, which is characterized by forms of necrosis and inflammation, including NLRP3 activation, elevated IL-1β serum concentrations[29], and local accumulation of immune cells[46, 47].

Genetically, ΔFHR1/3 protects against IgA nephropathy[16] and AMD[48, 49] and is a reported risk for atypical hemolytic uremic syndrome (aHUS)[50] and SLE[18], as well as rheumatoid arthritis[51] (all of which are associated with inflammation). Thus, it is tempting to speculate that in AMD and IgAN FHR1 is a major driver of inflammation and that ΔFHR1/3 protects against these diseases by reduced inflammation. The pathological mechanisms, however, underlying these diseases are more complex. That said, patients with these chronic diseases need to be evaluated to support the pro-inflammatory role of FHR1 and that symptoms arise from established lesions.

Most importantly, our data provide evidence that direct intervention with FHR1 function inhibits inflammation in human diseases like AAV to reduce progression and even induce regression. Therefore, targeting FHR1 or members of the FHR1 pathway like EMR2 presents as a promising therapeutic approach for future intervention to control inflammation.

## Methods

**Cell growth conditions**. Human umbilical vein endothelial cells (HUVEC, ATCC CRL-1730) were grown in DMEM (Lonza) with FBS (10%, PAA), ultraglutamine (1%, Lonza) and gentamicin sulfate (Lonza) at 37 °C and $CO_2$ (5%). Murine alveolar MH-S macrophages (ATCC CRL-2019) were grown in RPMI 1640 (Lonza) supplemented with heat-inactivated FBS (10% FBS), sodium-pyruvate (1 mM, Lonza), ultraglutamine (1%) and gentamicin sulfate (50 mg mL$^{-1}$) at 37 °C and $CO_2$ (5%). THP-1 cells (DSMZ, ATCC 16) were maintained in RPMI 1640 supplemented with FBS (10%), ultraglutamin (2 mM) and gentamicine sulfate at 37 °C and $CO_2$ (5%). U-937 cells (ATCC CRL-1593.2) were cultered in RPMI 1640 supplemented with FBS (10%) and gentamicin sulfate at 37 °C and $CO_2$ (5%).

Cells were authenticated and tested for mycoplasma contamination by ATCC, passaged every second day until passage 30. Normal human serum (NHS) was prepared from FHR1/3 sufficient as well as FHR1/3 deficient (ΔFHR1/3 NHS) blood derived from healthy volunteers, as determined by PCR[52] and Western blot analyses. After coagulation blood was centrifuged (10 min, 2000×$g$, 4 °C), and NHS kept frozen in aliquots at −80 °C. *C. albicans* cph1Δ/efg1Δ[53, 54], which cannot form hyphae, was grown overnight in YPD medium (D glucose (2%), peptone (1%), yeast extract in $H_2O$ (5%)) at room temperature.

**Isolation of human monocytes**. Biocoll (14 mL, Biochrom) was overlayed by a mixture of DPBS (5 mL, Lonza) and buffy coat (30 mL), derived from healthy male donors. After centrifugation (20 min, 550×$g$) the PBMC layer was washed with DPBS (5 min, 160×$g$). Biocoll and wash steps were repeated twice. Pellet was resuspended in IMDM (25 mL, Thermo Scientific) overlayed on Percoll (46%, GE Healthcare Life Sciences) in IMDM and centrifuged (20 min, 550×$g$). From PBMC layer monocytes were selected according to the protocol of the human Monocyte Isolation Kit II (MiltenyiBiotec). Monocyte experiments were performed in complete medium composed of 10% FBS and gentamicin sulfate in IMDM (Thermo Fisher Scientific).

**Isolation of murine bone marrow-derived macrophages**. All animal experiments were approved by the appropriate institutional and governmental committees for animal welfare and procedures conducted according to guidelines of the local Animal Use and Care Committee and the National Animal Welfare Laws (Zuchtrahmenantrag 02-05/16). Bone marrow cells were isolated from 10–19 weeks-old male mice (C57BL/6) and differentiated to macrophages according to the protocol of Francke et al.[55]. Blood was collected from 10–25 weeks old mice (C57BL/6) by cardiac puncture in hirudin S-Monovette (Sarstedt). Serum was collected by centrifugation (2000×$g$, 10 min, 4 °C) and stored at −20 °C.

**Proteins**. Recombinant FHR1, FHR1 SCR1-2, FHR1 SCR3-5, and FHR2 were expressed in *Pichia pastoris*[11, 56]. Recombinant FHL-1, FHR3, and FH SCR19-20 were expressed in baculovirus expression system[57-59]. BSA was purchased by AppliChem and factor H (FH) from Complement Technology. Full length FHRB (pCMV3-insert-Myc, Sino Biological Inc.) was restricted by Kpn1/Xba1, cloned into piCZαB (Invitrogen) expression vector and expressed in *P. pastoris*. Polyclonal rabbit FHRB antiserum was generated (Davids Biotechnologie GmbH). FHR2 dimerization mutant was generated by in vitro mutagenesis of the CFHR2 containing piCZα expression vector[60] using Quick Change Multi Site-Directed Mutagenesis Kit (Agilent Technologies) and the primer CAAGTTCCTACAGGG GAAGTTTTCTCTTACTACTGTGAGAGAATTTTGTGTCTCCTTCAAAATCC T. Mutagenesis was confirmed by sequencing. The His-tagged protein was expressed in *P. pastoris* strain X33 and purified by nickel chelate affinity chromatography. All proteins were evaluated by silver gel and western blot analyses.

**Cytokine experiments**. FHR1, FHR2, FHR3, FHL1, FH, and BSA (each 50 µL of 5 µg mL$^{-1}$) were coated on a High Binding ELISA plate (Sarstedt) for 1 h at 37 °C. Afterwards proteins were incubated with $1 \times 10^5$ monocytes/well in complete medium with or without 10% NHS and with or without stimulation with 5 ng mL$^{-1}$ LPS. After 20 h incubation at 37 °C and $CO_2$ (5%) cytokines in the supernatant were measured according to the protocol of the human cytokine ELISA kits: IL-1β, IL-6 and IL-10 (ThermoFisherScientific), IL-8 (Peprotech), IL-18 (R&D Systems) and TNFα (BioLegend). The absorbance was measured by microplate reader (TECAN, Safire2). For dose-dependent studies 50 µL of 0.6, 1.25, 2.5, and 5 µg mL$^{-1}$ of FHR1 were used for coating. For time dependent studies FHR1 or BSA were incubated with monocytes for 0, 15, 30, 60, 90, 120, 180, and 240 min. Fragments FHR1 SCR1-2, FHR1 SCR3-5 or FH SCR19-20 (each 50 µL of 5 µg mL$^{-1}$) were coated and incubated with monocytes in ΔFHR1/3 NHS (10%) or with LPS-treated monocytes in NHS (10%). FHRB (5 µg mL$^{-1}$) or DPBS were coated to a microtiter plate and incubated with mouse monocytes ($1 \times 10^5$) in normal mouse serum (10%) and 5% $CO_2$ with or without MCC950 (10 µM, InvivoGen) for 20 h at 37 °C. IL-1β was measured in the supernatant using a mouse IL-1β kit (BioLegend). Testing FHR1 activity in serum, plates were blocked with BSA prior incubation of monocytes with FHR1 (5 µg mL$^{-1}$) or BSA in serum with or without LPS. In another assay NHS was heat inactivated 30 min at 56 °C or inactivated by addition of EDTA (10 mM, Roth). To measure dose dependency, monocytes were incubated in 0, 0.1, 0.25, 0.5, 0.75, 1, 2.5, 5, 7.5, 10% NHS in IMDM without FBS.

For inhibition studies: C1q, C3, factor B and factor D depleted NHS was purchased by Complement Technology. C3 inhibition was performed with Compstatin (15 µM, Tocris Bioscience), C5 with Eculizumab (50 nM, Alexion Pharmaceuticals), CR3 by Simvastatin (20 µM, Sigma-Aldrich) and C3aR by

trifluoroacetate salt (TAS) (200 nM, Sigma-Aldrich). Classical and lectin pathways were inhibited with EGTA (10 mM, Sigma-Aldrich). NF-κB was blocked with BAY 11-7085 (30 μM, Sigma-Aldrich), TLR2 with pAb hTLR-2 (10 μg m$^{-1}$L, InvivoGen, LOT T2S-38-01), TLR4 with LPS-RS Ultrapure (22.2 μg mL$^{-1}$, InvivoGen), TLR6 with pAb hTLR-6 (10 μg mL$^{-1}$, InvivoGen, LOT T6S-36-01), Dectin 1 with WGP Soluble (1 μg mL$^{-1}$, InvivoGen), FcR with FcR Blocking Reagent (2 μL mL$^{-1}$, MiltenyiBiotec), CD14 with FITC hCD14 antibody (10 μg mL$^{-1}$, BioLegend, LOT B187006), CD36 with mAb hCD36 (5 μg mL$^{-1}$, Hycultec, LOT 17003M1114-B) and the human RAGE receptor with RAGE mouse anti-human clone 176902 (10 μg mL$^{-1}$, R&D Systems, LOT ICW0217021). The NLRP3-inflammasome was blocked with MCC950 (10 μM, Invivogen), Caspase-1 with VX-765 (50 μg mL$^{-1}$, InvivoGen), K$^+$ efflux with Glybenclamide (25 μg mL$^{-1}$, InvivoGen), Cathepsin B with Cathepsin B inhibitor (1 μM, Santa Cruz Biotechnology) and PLC with U73122 (10 μM, Abcam) or 1,10-Phenanthroline monohydrate (200 μM, AppliChem). Gβγ subunit of GPCR was blocked with Gallein (10 μM, Tocris) and EMR2 receptor via EMR2 antibody (10 μg mL$^{-1}$, R&D Systems, Cat. Nr. AF4894, LOT CCOS011805A). IL-1β was measured after 4 h incubation. The IL-1β response to FHR1 bound to C3b was determined after C3b (5 μg mL$^{-1}$) was coated for 1 h at 37 °C. C3b was blocked with blocking buffer I (200 μL) for 1 h at 37 °C and incubated with FHR1 (10 μg mL$^{-1}$) for 2 h at 37 °C. Each step was followed by washing four times with DPBS 0.05% Tween 20. Afterwards monocytes (1 × 10$^5$) were added and incubated for 20 h at 37 °C.

Healthy and necrotic (treated 35 min at 63 °C) HUVECs (each 1 × 10$^5$) were incubated with FHR1 or BSA (each 10 μg) for 20 min at 37 °C. After washing with DPBS (1% BSA), HUVECs (1.5 × 10$^4$) were incubated with monocytes (6 × 10$^4$) in CM with NHS (10%) for 20 h at 37 °C and CO$_2$ (5%). The supernatant was collected for IL-1β measurement. In addition, monocytes (1 × 10$^5$) were incubated with necrotic HUVECs (5 × 10$^4$) in NHS or ΔFHR1/3 NHS (each 10%) in CM. Afterwards ΔFHR1/3 NHS was reconstituted with FHR1 (0, 5, 10, 50, 75 or 100 μg mL$^{-1}$) or FH (10 or 100 μg mL$^{-1}$).

F(ab)$_2$ fragments of monoclonal FHR1 antibody[11] were generated with Pierce Mouse IgG1 Fab and F(ab)$_2$ micro Preparation Kit (Thermo Scientific) according to the provided protocol. Monocytes (1 × 10$^5$) were incubated with necrotic cells (5 × 10$^4$) and ΔFHR1/3 NHS (5%) or NHS (10%) and FcR blocker (2 μL mL$^{-1}$) with or without addition of F(ab)$_2$ (25 μg mL$^{-1}$) for 20 h at 37 °C.

To investigate the influence of FHR2, monocytes (1 × 10$^5$) were incubated with necrotic HUVECs (5 × 10$^4$) in ΔFHR1/3 NHS (5%) in CM with addition of BSA (3 μg) or FHR1 (3 μg) with or without FHR2 or FHR2DM (each 3 μg). FHR1 binding to low-density lipoprotein (LDL) and Malondialdehyde-modified LDL (MDA-LDL) (both Cell Biolabs) was performed by coating LDL or MDA-LDL (each 5 μg mL$^{-1}$) for 1 h at 37 °C, blocked with blocking buffer I (200 μL, AppliChem) for 1 h at 37 °C and incubated with FHR1 (10 μg mL$^{-1}$) for 2 h at 37 °C. Incubation with monocytes was followed by cytokine measurement as described above.

**RNA purification and analysis**. FHR1 (50 μL of 5 μg mL$^{-1}$) or BSA was immobilized and incubated with monocytes (1 × 10$^5$) in CM with or without NHS (10%) for 4 h at 37 °C. RNA was extracted using total RNA purification kit (NORGEN) and converted to cDNA by High Capacity RNA-to-cDNA Kit (Thermo Fisher Scientific). Quantitative PCR (StepOnePlus™ Real-Time PCR System, Thermo Fisher Scientific) was performed with cDNA derived from total RNA (0.25 ng according to NanoDrop ND-1000 Spectrophotometer (peqlab Biotechnologie), PerfeCTA SYBR Green Fast mix Low Rox (Quanta Bio) and primers ACTB forward 5′GCTAAGTCCTGCCCCTCATTT′3, ACTB reverse 5′GTACAGGTCTTTG CGGATGT′3, IL-1β forward CTCTCACCTCTCCTACTCACTT′3, IL-1β reverse TCAGAATGTGGGAGCGAATG′3, and TNFα forward 5′CCAGGGACCTCTCT CTAATCA′3, and TNFα reverse 5′TCAGCTTGAGGGTTTGCTAC′3. Actin beta was used as endogenous control. Data were analyzed with Expression Suite Software version 1.1 and StepOne™ Software v2.3. RNA from FHR1 treated monocytes from four different donors were mixed (total 3.4 μg) and sequenced by LC Sciences. Data were analyzed using DAVID 6.7 and Panther 9.0 software.

**LDH measurement and cell titer blue assay**. Immobilized FHR1 (50 μL of 5 μg mL$^{-1}$) in triplicate was incubated with monocytes (1 × 10$^5$) in IMDM with NHS (0.25%) for 20 h at 37 °C in CO$_2$ (5%). For LDH measurement, after 19 h lysis buffer (10 μL) was added for 45 min at 37 °C. Supernatants (50 μL) were incubated with the reaction mixture of Pierce LDH Cytotoxicity Assay Kit (50 μL, Thermo Scientific) for 30 min at RT. The reaction was stopped with stop solution (50 μL) and absorbance was measured at 490 nm. In the cell titer blue (CTB) assay, supernatant was removed after 20 h, cells washed and incubated with 100 μl CTB solution (Promega) for 6 h at 37 °C and absorbance was measured at 570–600 nm. To induce pyroptosis monocytes were treated with Nigericin (20 μM, Santa Cruz Biotechnology) for 20 h or Triton-X-100 (0,1%, Sigma-Aldrich) for 3 min.

**ROS measurement**. FHR1 (50 μL of 5 μg mL$^{-1}$), BSA or DPBS were coated to a microtiter plate for 1 h at 37 °C. Proteins were incubated with monocytes (1 × 10$^5$) in CM with or without 10% NHS. As positive control monocytes (1 × 10$^5$) were incubated with C. albicans cph1Δ/efg1Δ (5 × 10$^4$). After 20 h incubation at 37 °C

and CO$_2$ (5%) CellROX™ Deep Red Reagent (2 μL mL$^{-1}$, Thermo Scientific) was added and the absorbance measured at Ex/Em 480/515-550 nm.

**Protein binding to HUVECs**. Binding of FHR1 to healthy and necrotic HUVECs (each 3 × 10$^5$) was determined by incubation with FHR1, FH or BSA (each 10 μg) for 20 min at 37 °C in DPBS (1% BSA). Cells were stained with mAb C02 (1:500)[61] and Alexa Fluor 488 donkey anti-mouse IgG (1:500, Invitrogen, LOT 1820538) with Viability Dye eFluor® 780 (1:10000, eBioscience), each for 20 min at 4 °C in DPBS (1% BSA). Between each step the cells were washed two times with DPBS (1% BSA) and fluorescence was measured via flow cytometry (gated according to FSC/SSC).

FHR1 fragments SCR1-2 and SCR 3-5 (each 50 μl of 60 μg mL$^{-1}$) were incubated with necrotic cells (5 × 10$^5$) for 30 min at 37 °C in DPBS (1% BSA). The cells were stained using α-Penta His (1:500, QUIAGEN) and Alexa 488 donkey α-mouse (1:500, Invitrogen, LOT 1820538), Viability Dye eFluor® 780 (1:10.000) each for 20 min at 4 °C in DPBS (1% BSA). Between each step cells were washed two times with DPBS (1% BSA) and fluorescence measured by flow cytometry (gated according to FSC/SSC).

Binding of FHR1 and FH from NHS was evaluated by incubating healthy or necrotic HUVECs (each 1 × 10$^6$) in 50% NHS (diluted in DPBS) for 30 min at 37 °C. After extensive washing, cells were lysed 30 min on ice in lysis buffer (70 μL) pH 7.4 with Triton X-100 (1% v/v) (Sigma) and Protease Inhibitor Cocktail (Roche) in DPBS. Roti®-Load 2 (Roth) 13.5 μL was added to each sample (40 μL) and heated for 10 min at 95 °C. Samples (40 μl) were separated via SDS-PAGE and immunoblotted using polyclonal FHR1 antiserum[62] (1:1.000) and HRP-conjugated α-rabbit antibody (1:1000, Dako, LOT 00087384). FHR1 (10 μg mL$^{-1}$) was again bound to necrotic HUVECs (45 min at 65 °C) for 30 min at 37 °C and cells were stained with polyclonal FHR1 antiserum[62] (1:1.000) for 30 min at RT. Cells were incubated with Alexa FluorTM488 donkey anti-rabbit IgG (1:500, Invitrogen, LOT 1796375), DAPI (10 μg mL$^{-1}$, Sigma), and WGA Texas Red (1:100, Invitrogen) for 30 min at RT. Between each step, cells were washed twice with 1% BSA in DPBS. Images were captured using LSM 710 (Zeiss) with ZEN 2009 software. Uncropped blot is provided in the Source Data file.

**Protein binding to necrotic-type cells**. Apoptosis was induced by treatment of HUVEC cells with staurosporine (0,5 μg ml$^{-1}$, o. n., Sigma-Aldrich). Afterwards cells (4.5 × 10$^5$) were incubated with BSA, FHR1 (20 μg ml$^{-1}$) or NHS (50%) for 20 min at 37 °C. The cells were stained using a monoclonal mouse FHR1 antibody[11] (1:500), followed by Alexa 488 donkey α-mouse (1:500, Invitrogen, LOT 1820538), Viability Dye eFluor® 780 (1:10.000) and FITC Annexin V (1:20, Bio-Legend, LOT B223857) each for 20 min at 4 °C. Necroptosis was induced by treatment of PMA preincubated (1 μg ml$^{-1}$, o. n., Sigma-Aldrich) U937 cells with Z-VAD-FMK (50 μM, Santa Cruz Biotechnology) for 1 h at 37 °C, followed by TNFα (50 ng ml$^{-1}$, gibco) and Cycloheximide (50 μg ml$^{-1}$, biomol) for 4 h 20 min at 37 °C. Pyroptosis was induced by treatment of PMA preincubated THP1 cells with Nigericin sodium salt (20 μM, Santa Cruz Biotechnology) for 2 h 30 min at 37 °C. Afterwards necroptotic cells (3.5 × 10$^5$) or pyroptotic cells (3 × 10$^5$) were incubated with BSA or FHR1 (50 μg ml$^{-1}$) for 30 min at 37 °C and stained with mAb C02 (1:500)[61] and Alexa Fluor 647 anti-mouse IgG (1:500, Cell Signaling Technology, LOT 14) with Viability Dye eFluor® 780 (1:10.000) and FITC Annexin V (1:20, BioLegend, LOT B223857) each for 20 min at 4 °C in DPBS (1% BSA). Between each step the cells were washed two times with DPBS (1% BSA) and fluorescence was measured via flow cytometry (gated according to FSC/SSC and VD/Annexin).

**Proximity ligation assay**. FHR1 (20 μg mL$^{-1}$) was incubated with necrotic HUVECs (45 min at 65°) at 37 °C for 60 min with shaking at 700 rpm. Washed cells (5 × 10$^5$) were seeded onto Poly-L-lysine (Sigma-Aldrich) coated diagnostic slides, type PTFE (Carl Roth) and incubated with polyclonal rabbit FHR1 anti-serum[62] (1:500) and monoclonal mouse MDA-LDL antibody E014[63] (1:100). Control cells were incubated with polyclonal rabbit FHR1 antiserum[62] (1:500) and mouse IgM κ Isotype Ctrl antibody (Biolegend, LOT B214463) (1:50). PLA assay was performed according to manufacturer's protocol provided with Duolink In Situ Red Starter Kit Mouse/Rabbit (Sigma-Aldrich). Images were captured using LSM 710 with ZEN 2011.

**FHRB binding to mouse alveolar macrophages**. Healthy as well as necrotic (30 min at 63 °C) mouse alveolar macrophages (each 5 × 10$^5$) were incubated with FHRB or BSA (each 10 μg) for 20 min at 37 °C followed by rabbit FHRB antiserum, generated to the recombinant protein (1:500, Supplementary Fig. 1a) and Alexa 647 goat α-rabbit (1:500, Invitrogen, LOT 1893813) together with Viability Dye eFluor® 780 (1:10.000), each for 20 min at 4 °C. Between each step cells were washed two times in DPBS with 1% BSA. FHRB binding was analyzed by flow cytometry (gated according to FSC/SSC).

**FHR1 binding to lipids and proteins**. LDL, MDA-LDL (both Cell Biolabs) or BSA (each 5 μg mL$^{-1}$) were coated for 1 h at 37 °C, blocked with 200 μL blocking buffer I for 1 h at 37 °C and incubated with 30 μg mL$^{-1}$ FHR1 2 h at RT, polyclonal FHR1 antiserum[11] (1:1000) in Cross Down buffer (AppliChem) as well as HRP-labeled

polyclonal anti-rabbit IgGs (Dako, LOT 00087384) 1 h at RT. All steps included 3–5 wash steps with DPBS 0.05% Tween 20 (Sigma-Aldrich). Afterwards TMB (50 µL, eBioScience) was added and reaction was stopped with $H_2SO_4$ (2 M, Roth). The absorbance at 450 nm was measured with a microplate reader. MDA-LDL ($5 \mu g mL^{-1}$) was coated and after blocking FHR1 (0, 1, 2.5, 5 or $10 \mu g mL^{-1}$) were incubated for 2 h 37 °C. The kinetics of FHR1 binding to MDA-LDL ($5 \mu g mL^{-1}$) were measured via surface plasmon resonance technique using (Biacore 3000). MDA-LDL was immobilized to a sensor chip (CM5) using a standard amine-coupling and FHR1 (250 – 2.000 nM) or FHR1 SCR 1-2 or SCR 3–5 (each 10 µg $mL^{-1}$) in PBS injected with a flow rate of $5 \mu L min^{-1}$ at 25 °C.

FHR1 binding to C3b was analyzed by coating C3b or FHR1 (each $5 \mu g mL^{-1}$) to a microtiter plate for 1 h at 37 °C, blocked with blocking buffer I (200 µL) for 1 h 37 °C and incubated with FHR1 ($10 \mu g mL^{-1}$) for 2 h at 37 °C. The ELISA was continued as described for MDA-LDL. Binding of recombinant human EMR2 to FHR1 was determined by immobilizing FHR1 ($10 \mu g mL^{-1}$) or gelatin to a microtiter plate o. n. at 4 °C. Proteins were blocked with blocking buffer for 1 h at 37 °C, incubated with EMR2 (R&D Systems) or BSA ($30 \mu g mL^{-1}$) for 2 h at 37 °C, and binding detected with polyclonal EMR2 antiserum ($0.5 \mu g mL^{-1}$, R&D Systems, LOT CCOS011807A) in Cross Down Buffer and HRP-labeled polyclonal anti-sheep IgGs (Dako, LOT 00082910), each 1 h at RT. All steps included 3–5 wash steps with DPBS 0.05% Tween 20. Afterwards TMB (50 µL, eBioScience) was added and reaction was stopped with $H_2SO_4$ (2 M, Roth). The kinetics of FHR1 binding to EMR2 was measured via surface plasmon resonance technique using a Biacore 3000 instrument as described above. Different concentrations of FHR1 (125–2.000 nM) in PBS were used.

**Ex vivo whole blood model**. Blood was collected from healthy volunteers with FHR1 and ΔFHR1/3. Serum was separated from blood by centrifugation (2.000 × g, 10 min). RBCs were removed from whole blood by water lysis and centrifugation (2000×g, 10 min). Whole blood cells ($7.5 \times 10^5$) were added to 12 mm Poly-l-lysine coated coverslips in a 24-well plate in RPMI medium (Lonza) supplemented with heat-inactivated FCS (10%) and ultra-glutamine 1 (2 mM, Lonza) and incubated for 2 h at 37 °C and 5% $CO_2$ to allow to adhere to the coverslips. Other whole blood cells ($7.5 \times 10^5$) from the same donor were incubated at 65 °C for 45 min to generate necrotic cells. The necrotic cells were incubated in NHS (100%) from the same donor and collected by centrifugation (2.000×g). Necrotic cells were resuspended in medium along with 10% NHS and were added to coverslips for adhering to the surface. Whole blood cells, necrotic cells and NHS from one donor were incubated for 1.5 h and reaction was stopped by addition of paraformaldehyde (4%, Roth). Cells were permeabilized with saponin (0.1%, Sigma-Aldrich). Fc receptors of the cells were blocked with FcR Blocking Reagent (Miltenyi Biotec) and Duolink blocking solution (Sigma-Aldrich). Cells were then treated with monoclonal mouse FHR1 antibody[11] (1:200) and polyclonal rabbit anti-EMR2 antiserum (Abcam, LOT GR152070-7) (1:200) or TGFβRI antibody (Sigma Aldrich, 211126) (1:200). PLA assay was performed according to manufacturer's protocol provided with Duolink In Situ Red Starter Kit Mouse/Rabbit (Sigma-Aldrich). Images were captured using LSM 710 equipped with ZEN 2011. Fluorescent intensity of the images was calculated using Fiji software.

**Replacement of FHR1 by FHR2 or FHR2DM**. FHR1 SCR 1–2 ($5 \mu g mL^{-1}$) was immobilized on a microtiter plate over night at 4 °C, incubated with NHS (5%) in DPBS for 1 h at 37 °C, and FHR2 or FHR2DM (0, 1, 5, or $15 \mu g mL^{-1}$) was added to each probe. FHR1 was detected by incubation with monoclonal antibody C18[61] (1:1000) for 1 h at RT and FHR2DM with monoclonal FHR2 antibody[60] and HRP-conjugated anti-mouse IgG (1:1000, Dako, LOT 20013032). All steps included 3–5 wash steps with DPBS, 0.05% Tween 20.

**Patients**. Patients with ANCA-associated necrotizing glomerulonephritis ($n = 314$) as diagnosed by kidney biopsies and PR3 and MPO antibodies and AS patients with acute coronary artery disease awaiting coronary artery bypass surgery were included in the cohort study. Blood and tissue samples from patients were collected and analyzed in this study. In addition, blood samples from healthy donors ($n = 55$) were collected as controls.

**Immunohistochemistry**. Atherosclerotic plaques were obtained from patients with high-grade aortic sclerosis with concomitant ascending aorta replacement during coronary artery bypass grafting. Atherosclerotic plaques consisted mainly of the diseased area resulting from the surgical excision. Kidney biopsies were taken from AAV patients. Human coronary plaques and kidney biopsies were segmented in blocks of 3–4 mm, fixed in formalin overnight, decalcified in 0.5 M EDTA (pH 7.2) and embedded in paraffin (FFPE)[64]. Tissue sections (1 µm) were stained for presence of FHR1, C3c, and FHR2 by pre-incubating the sections with a protease solution for 30 min at 40 °C. Subsequently the sections were incubated with monoclonal FHR1 antibody[11] (1:40.000), C3c antibody (1:3.000, Dako, LOT 00055806) or monoclonal FHR2 antibody[60] overnight at 4 °C. As detection antibodies AffiniPure Rabbit Anti-Mouse IgG (H + L) (Jackson ImmunoResearch, LOT 136143) and Alkaline Phosphatase (APAAP complex) (ALPL) (Monoclonal Antibody MyBioSource, LOT 2E14317) were used. Stained slides were developed using ZytochemPlus/POLAP100 (Zytomed),

incubated with polymer 1 and 2 and rinsed in between with PBS. Slides were stained in fuchsin naphthol As-Bi phosphate substrate mixture for 30 min and for nuclear counterstaining in hemalaun (Mayer) for 1 min and quantified using ImageJ or ZEN software. For PLA assays Paraffin-embedded sections of human atherosclerotic plaques or AAV kidney biopsies were deparaffinized by placing them consecutively in Roticlear (Carl Roth), 100% ethanol, and 95% ethanol. Sectioned tissues were boiled in 10 mM Na citrate buffer (pH 6.5) for antigen retrieval and blocked with normal serum block (Biolegend) and FcR Blocking Reagent diluted in antibody diluent (Roth). Tissues were then treated with monoclonal mouse anti-FHR1 antibodies[11] (1:200) and polyclonal rabbit anti-EMR2 antiserum (Abcam, LOT GR152070-7) (1:200) or TGFβRI antibody (Sigma Aldrich, 211126) (1:200) diluted in antibody diluent. For antibody control tissues were treated with exclusively rabbit anti-EMR2 antiserum (1:200). PLA assay was performed according to the manufacturer's protocol provided with Duolink In Situ Red Starter Kit Mouse/Rabbit (Sigma-Aldrich). Images were captured using LSM 710 equipped with ZEN 2011.

**Analysis of serum samples**. IL-1β serum levels were determined by high sensitive IL-1ß Kit (Thermo Fisher Scientific), anti-MPO-antibodies (ORG 519 Orgentec), anti-PR3-antibodies (ORG 618, Orgentec) by ELISA, and FHR1 concentration by Human CFHR1 ELISA Kit (RayBiotech) according to the protocol provided by the manufacturers. CRP levels were measured by module C701 (Cobas8000). All samples below CRP detection limit of $5 mg dL^{-1}$ were defined as $5 mg dL^{-1}$. Presence of FHR1 in NHS samples was determined by Western Blot analysis and absence confirmed by PCR[19]. Roti®-Load 2 (2.5 µL, Roth) was added to NHS (1 µL) diluted in DPBS (10 µL) and samples were separated by SDS-PAGE. Immunoblotting was performed with polyclonal FH antiserum (1:7.500, CompTech, LOT #4) and HRP-conjugated goat antibody (1:2.500, Dako, LOT 00073984).

**Ethics**. After informed consent was obtained, patient data, blood, and tissue samples were collected according to the guidelines of the local ethics committees (PV3162, PV4068, and PV5657 University Hospital Eppendorf, Hamburg and 5071-02/17 Friedrich-Schiller University, Jena) and the Guidelines of the World Medical Association Declaration of Helsinki. Patient's data are summarized in Supplementary Table 2.

**Statistical analysis**. Significant differences between two groups were analyzed using the unpaired two-tailed Student's $t$-test of GraphPad Prism 5 for Windows. AAV and AS patients were analyzed using the unpaired two-tailed $t$-test with Welch's correction and Kruskal–Wallis Test. Semi quantitative IHC was analyzed using Wilcoxon Signed Rank test. Correlation between IL-1β/CRP, IL-1β/ANCA, IL-1β/FHR1, and FHR1/eGFR were analyzed using Spearman correlation test. Values of $^*p \le 0.05$, $^{**}p \le 0.01$, $^{***}p \le 0.001$ were considered as statistically significant.

**Reporting summary**. Further information on research design is available in the Nature Research Reporting Summary linked to this article.

## Data availability
The data generated for this study have been deposited at the Gene Expression Omnibus (GEO) under accession code GSE119025. The source data underlying Figs. 1–3, 4f–h, 5a, b,d–j, 6a–c,e,f, 7, and Supplementary Figs. 1, 2, 3b–d, f, h, and 4d–m including all uncropped gels and blots are provided as a Source Data file.

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

## Acknowledgements

This work was supported by the Deutsche Forschungsgemeinschaft (DFG, Sk 46), within Collaborative Research Center CRC124 FungiNet Project C4 (C.S., B.J.) and C6 (P.Z.) and SFB1192 (R.A.K.S., T.W., P.F.Z). S.I. is doctoral researcher at the Jena School for

Microbial Communications (JSMC) in Jena, and L.D.H. at the International Leibniz Research School (ILRS), Jena, Germany. We thank Ina Löschmann, Monika von der Heide, Andrea Hartmann and Steffi Hälbich for the excellent technical support.

## Author contributions

C.S. and S.I. designed and supervised the study, wrote the paper, and designed Figs. 4i and 8. All authors discussed the data and contributed to the manuscript. S.I., L.D.H., S. M., S.W. performed experiments. S.L.H.Z., S.R.B., R.A.K.S., E.G., H.R., and T.W. collected patient material, performed IHC and characterized patient cohorts. B.J. provided mouse material. P.F.Z. provided additional comments and ideas to finalize the paper.

## Additional information

**Competing interests:** S.I., P.F.Z., and C.S. have filed a patent (DE10 2018 120 016.1). The remaining authors declare no competing interests.

