## [Peer Review File · Nature Communications]

Reviewer expertise:

Reviewer #1: AAV

Reviewer #2: complement

Reviewer #3: inflammatory cell death

Reviewer #4: atherosclerosis

Reviewers' comments:

Reviewer #1 (Remarks to the Author):

The authors describe an interesting study in which they identify a novel potentially pro-inflammatory function for complement factor H-related protein 1 (FHR1), together with studies of patients, predominantly with ANCA-associated vasculitis (AAV), from which they conclude the data: " provide evidence that direct intervention with FHR1 function inhibits inflammation in human diseases like AAV to reduce progression and even induce regression".

Specifically they show: (i) immobilised but not fluid phase FHR1, stimulates human monocytes to release interleukin-1beta (IL-1beta) and other pro-inflammatory cytokines (either alone or in the presence of LPS), whilst attenuating release of the anti-inflammatory cytokine IL-10; (ii) the effect is serum dependent but complement independent and mediated by binding of FHR1 to EMR2 on the monocyte surface which in turn activates the NLR3 inflammasome; (iii) the FHR1 N-terminus SCR 1-2 domains bind MDA in damaged lipids on necrotic but not healthy human umbilical endothelial cells (HUVEC) and activate human monocytes, whereas neither FHR1 nor FH bound to cell surface C3b via C terminus SCR 3-5 evoke a monocyte response; and (iv) FHR1 binding to necrotic cells is inhibited by FHR2 with which it forms heterodimers. Collectively these data are highly convincing and the FHR1 function identified is novel and potentially important.

In the second part of the study, the authors investigate the clinical relevance of the FHR1/necrotic cell/monocyte interaction using AAV and also atherosclerosis as models: (i) using immunohistology they show FHR1 is deposited in glomeruli with segmental necrosis in patients with AAV, and in atheroma plaques from aortas; (ii) identify "FHR1/EMR2" complexes in nephritic glomeruli in AAV with the PLA; (iii) show in patients with AAV that circulating IL-beta and C-reactive protein concentrations and the proportions of glomeruli with segmental necrosis and crescents are significantly lower in those who are FHR1 deficient (n=11) than in FHR1 sufficient individuals (n=314). These studies (described in Figure 7 of the manuscript) clearly show FHR1 is deposited at sites of injury and provide a prima facie case that it may influence its severity injury. However the data as described are over interpreted and some additional specificity controls are needed. Presently, they do not justify the authors conclusions that "FHR1 binds to necrotic cells in vivo", and "Altogether the data demonstrate an in vivo influence of FHR1 on inflammation."

Specific comments

1. The authors should confirm the validity of the proximity ligation assay (PLA) to assess specific binding of FHR1 to EMR2 with an additional control experiment using antibodies to FHR1 together with antibodies specific for another protein monocyte surface to ensure that the FHR1/EMR2 pairing is specific rather than reflecting the proximity of FHR1 to an adjacent protein on the monocyte surface.
2. Are the necrotic cells linked to monocytes by FHR1 internalised.
3. It would be interesting to know whether FHR1 bound to necrotic cells or whether it also bound apoptotic ones, and if so whether the induces pro-inflammatory monocyte responses or abrogates the anti-inflammatory response they normally evoke.

4. It is an over-interpretation to state that Figure 7a shows that “FHR1 attaches to necrotic cells surrounding fresh fibrinoid necrotic lesions in the glomerulus”. The present images merely show that FHR1 is deposited in necrotic lesions – in common with all other deposited complement proteins the mechanism of deposition/retention in the glomerulus cannot be ascertained from these light microscopy images. There are many other reasons why FHR1 could be deposited including by binding FH or C3b, dimerising with FHR2 etc, etc. Accordingly, it would be valuable to know which other complement components were deposited on the same location, including FH, FHR2 and C3. The absence of binding in FHR1 deficient individuals merely tells one that the antibody used was specific and informative about the biology of FHR1, although in concert with staining for other complement proteins it could be.

5. Figure 7c the authors use the PLA to identify EMR2/FHR1 complexes renal biopsies from patients with AAV. These clearly demonstrate a signal in FHR1 sufficient individuals but not in deficient ones (Δ FHR1/3). This is a good negative control but not one for specificity: this requires substituting the anti-EMR antibody for one specific for a different monocyte surface receptor.

6. Figures 7 d and e show that the IL-1beta and CRP concentrations are significantly higher in FHR1 sufficient individuals with AAV than Δ FHR1/3 ones but there is considerable overlap between the two groups that differ greatly in size. Accordingly, the results need to be interpreted with extreme caution, not least because of potential ascertainment biases caused by the small number of Δ FHR1/3 individuals studied and possible differences in the stage of the disease or concurrent treatment. More clinical data should be included to mitigate the possibility of ascertainment bias.

7. Figure f and g demonstrate the Δ FHR1/3 individuals have significantly lower proportion of crescents and segmental necrotic lesions. Again more clinical data and a validation cohort are required before definitive conclusions can be made about the possible direct effect of FHR1. Equally important, even a direct effect could be due to FHR1 functions other than its binding necrotic cells; for example, its reported direct inhibitory effects of C3 and C5 convertases (the latter especially pertinent because of the therapeutic efficacy of inhibition of the C5a receptor in nephritis in AAV). Similar comments apply to reported relapses in Δ FHR1/3 individuals.

8. Of interest, the authors cite the repeatedly validated protective effect of in Δ FHR1/3 on susceptibility to another type of glomerulonephritis, mesangial IgA disease. Recent analyses of large cohorts have shown no detectable effect on severity of IgA nephritis, which contrasts with the authors’ suggestion of the opposite in AAV in which the major effect is on severity and/or progression. Data on as yet unreported effects of in Δ FHR1/3 on genetic susceptibility should be available in the public datasets from the European and North American GWASs published over the past.

In summary, it is important to address the clinical issues raised. In some cases this will require new experiments or additional data but in many cases they could be remedied with by drawing more conclusions limited to the data. Perhaps changes to the introduction to set out the context succinctly might also help.

Andrew Rees

Reviewer #2 (Remarks to the Author):

This paper by Irmscher et al describes a comprehensive sets of experiments to support the central hypothesis that FHR1, or factor H related protein 1, plays a role in sterile inflammation by binding to necrotic cells and triggering EMR2 activation and signaling, ultimately to inflammasome NLRP3 induction and cytokine production (IL-1-beta and TNF-alpha etc). The study is quite mechanistic and

of considerable depth, employing a number of approaches and correlates, including dissection of signaling pathways, RNA seq analysis of gene regulation in FHR1-stimulated monocytes and human patient population analysis). The findings and conclusions are significant and could potentially explain why FHR1/3 gene deficiency is protective in AMD and IgAN. There are however a number of issues remain to be addressed and clarified.

1. The authors presented data showing serum is required for the FHR1 activity and the serum element is heat-sensitive, but there is no information or discussion about why serum is required and what are the essential serum element(s) and how it works. The authors have excluded complement being the required heat-sensitive element.

2. FH 18/19/20 domains are highly homologous to SCR3-5 of FHR1 and it was shown FH 19-20 can do the same thing as FHR1. Can whole FH do the same thing as FHR1, i.e binding to necrotic cells and triggering cytokine production in monocytes?

3. While the data and hypothesis presented could potentially explain why FHR1/FHR3 gene deletion is protective for AMD, IgAN and ANCA, the opposite association, i.e. FHR1/FHR3 deficiency increases the risk of SLE, aHUS and RA cannot be explained. What's are the author's thoughts on the latter?

4. page 19, line 366-367, why FHR1 antibodies would augment rather than reduce the inflammatory response in NHS? Similarly on the same page, line 369-370, "In conclusion, inhibition of the FHR1 N-terminus augmented FHR1 induced inflammation...." seems to contradict the sentence immediately before it which stated " Monoclonal FHR1 F(ab)2reduced the IL-1b release....by about 50%.." This paragraph is very confusing

Reviewer #3 (Remarks to the Author):

In their paper, Irmscher, Brix and Zipfel et al. present data on necrotic cell-induced inflammation. They demonstrate that FHR1 specifically induces NLRP3-mediated IL-1 β maturation and secretion when immobilized. They further suggest that FHR1 binds to EMR2 (ADGRE2), mutations in which are associated with bacteria-induced urinary tract infections, the defense against which obviously involves necrotic type cell death. They use IF to detect the FHR1/EMR2 complex on the surface of what they refer to as necrotic cells. They confirm a previously described MDA-LDL binding of FHR1 (published in Weismann et al., nature 2011, ref 17). Finally, they study FHR1 in samples from AAV and AS patients. Unfortunately, the study is entirely correlative and does not answer the major questions raised by the correlative data, e.g. how FHR1 drives inflammasome assembly or how necrotic cells in tissues might specifically bind to this factor. The role of EMR2 is touched upon in a very superficial manner. Methodologically, several severe issues must be raised (see below), such as the induction of necrosis, the methods to evaluate the inflammasomes and the detection of necrosis in tissue samples including the lack of validation of antibodies in knockout mice. In addition, the authors ignore the readily available mouse models of AAV which would be very useful for mechanistic insights in this study. Importantly, the cell culture models of necrosis-induction are highly artificial (heat stress only). To be more convincing, a method of regulated necrosis, e.g. pyroptosis, should have been chosen for the investigation of NLRP3-mediated necrosis (as obviously present in some other figures). In summary, this is an interesting but entirely correlative study, performed with limited expertise to the field and poor methodology. I cannot recommend publication in Nature Communications.

Major concerns

- How does FHR1 drive inflammasome activation? This requires the analysis of inflammasome components (ASC, NLRP3, caspase-1 cleavage), but also detection of downstream markers and necrosis (pyroptosis) mediated by GSDMD.
- How does EMR2 regulate pyroptosis signalling? The EMR2 receptor undergoes autocatalytic cleavage, producing an extracellular subunit that noncovalently binds a transmembrane subunit. Is this

signalling involved in inflammasome activation, ASC-Spec formation, caspase-1/11 activity and GSDMD-cleavage? Is GSDMD required for IL-1 β release? A simple crKO of EMR2 would help to link or unlink the underlying mechanisms here. However, a cleavage deficient mutant knock-in would be required for mechanistic links to be concluded.

- There are mouse models available and widely used for the study of AAV. How would a EMR2-ko/cleavage-deficient mutant and/or an FHR1 (if present in mice)-ko respond to a model like this?
- Interestingly, the authors, as much as they represent experts in the field of complement, appear to ignore the literature on necrosis and inflammasomes. They ignore the obvious connection to gasdermin D/pyroptosis despite several high impact papers were published in the recent 3 years on this topic. They also do not appear to be aware of the novel concepts on necroinflammation which would be critical if the title remains unchanged.
- Induction of necrosis is performed by simply heating the cells to 63°C. This is a highly artificial model. Both AAV and AS have been demonstrated to involve processes of regulated necrosis (pyroptosis, necroptosis, parthanatos and others). Why do the authors work on artificial necrosis instead of inducing pyroptosis or necroptosis in their systems. These pathways may be associated with a different immune response/DAMP release compared to simply heat stress. Especially when putting IL-1b on the central platform in Fig. 1 and 2, why not work on pyroptosis induced by e.g. nigericin?
- How do the authors explain the upregulation of TNF α and IL-6 in Fig. 1j and 1k? Are these simply elements of a cytokine storm or is this a highly specific upstream signal from necroptotically dying cells? If the latter is the case, necroptosis should be studied in the presence and absence of FHR1.

Minor remarks

- The discussion and introduction should include the recent literature/concepts on pyroptosis (GSDM-mediated necrosis) and its influences on the immune system.
- The resolution of PLA is insufficient. This is a complex technique that should be used in addition to others only if very good primary antibodies are available which does not seem to be the case here.
- Fig. S4 – Fig. S4A does not allow any conclusion in the presented quality. How representative is the western blot in Fig. S4G?

Reviewer #4 (Remarks to the Author):

Irmscher S et al identify a new receptor for FHR1. Moreover, they observed that FHR1 binds to necrotic cells to exacerbate pro-inflammatory cytokine production/release in monocytes. This work is important and describes a mechanism associated with persistent inflammation in disease like atherosclerosis and AAV, where the accumulation of necrotic cells is a critical pathological factor. I think this body of work is thorough and I truly enjoyed learning from this manuscript. I only have minor comments.

Comments:

1) Excellent aspects about the work:

--This work focuses on human cells and human tissues therefore the translational relevance is very clear.

--The authors use RNA seq and functional in vitro assays to help determine a new receptor for FHR1 on monocytes.

2) Please revise the statistics throughout. In several cases two-tailed t test is not the best or most accurate statistical method for your data.

3) Given that apoptotic cells are also present in atherosclerosis and AAV do you know if FHR1 binds to apoptotic cells? Apoptotic cells do not evoke inflammation and exert a tissue reparative response on

monocytes/macrophages. If FHR1 does not bind to ACs, then your new mechanism may also describe why necrotic cells evoke pro-inflammation on monocytes and why ACs do not.

4) the necrotic cells were generated via heat-stress. Do necroptotic cells also bind FHR1? Do pyroptotic cells bind FHR1?

5) Do FHR1 knockout mice exist? Doesn't appear so, based on a cursory search. If they become available in the future, then this work will lay the foundation for causative experiments in which one can test whether the lack of FHR1 can diminish necrotic cores in atherosclerosis and any disease where necrosis underpins the pathology.

Reviewer #1 (Remarks to the Author):

The authors describe an interesting study in which they identify a novel potentially pro-inflammatory function for complement factor H-related protein 1 (FHR1), together with studies of patients, predominantly with ANCA-associated vasculitis (AAV), from which they conclude the data: “ provide evidence that direct intervention with FHR1 function inhibits inflammation in human diseases like AAV to reduce progression and even induce regression”.

Specifically they show: (i) immobilised but not fluid phase FHR1, stimulates human monocytes to release interleukin-1beta (IL-1beta) and other pro-inflammatory cytokines (either alone or in the presence of LPS), whilst attenuating release of the anti-inflammatory cytokine IL-10; (ii) the effect is serum dependent but complement independent and mediated by binding of FHR1 to EMR2 on the monocyte surface which in turn activates the NLR3 inflammasome; (iii) the FHR1 N-terminus SCR 1-2 domains bind MDA in damaged lipids on necrotic but not healthy human umbilical endothelial cells (HUVEC) and activate human monocytes, whereas neither FHR1 nor FH bound to cell surface C3b via C terminus SCR 3-5 evoke a monocyte response; and (iv) FHR1 binding to necrotic cells is inhibited by FHR2 with which it forms heterodimers. Collectively these data are highly convincing and the FHR1 function identified is novel and potentially important.

In the second part of the study, the authors investigate the clinical relevance of the FHR1/necrotic cell/monocyte interaction using AAV and also atherosclerosis as models: (i) using immunohistology they show FHR1 is deposited in glomeruli with segmental necrosis in patients with AAV, and in atheroma plaques from aortas; (ii) identify “FHR1/EMR2” complexes in nephritic glomeruli in AAV with the PLA; (iii) show in patients with AAV that circulating IL-1beta and C-reactive protein concentrations and the proportions of glomeruli with segmental necrosis and crescents are significantly lower in those who are FHR1 deficient (n=11) than in FHR1 sufficient individuals (n=314). These studies (described in Figure 7 of the manuscript) clearly show FHR1 is deposited at sites of injury and provide a prima facie case that it may influence its severity injury. However the data as described are over interpreted and some additional specificity controls are needed. Presently, they do not justify the authors conclusions that “FHR1 binds to necrotic cells in vivo”, and “Altogether the data demonstrate an in vivo influence of FHR1 on inflammation.”

Specific comments

1. The authors should confirm the validity of the proximity ligation assay (PLA) to assess specific binding of FHR1 to EMR2 with an additional control experiment using antibodies to FHR1 together with antibodies specific for another protein monocyte

surface to ensure that the FHR1/EMR2 pairing is specific rather than reflecting the proximity of FHR1 to an adjacent protein on the monocyte surface.

Response

In our manuscript we showed that FHR1 binds to EMR2 and that the FHR1 induced inflammatory reaction in monocytes is inhibited by blocking EMR2. In addition, we demonstrate in two different staining experiments -on necrotic cells with monocytes and in tissue sections- that FHR1 binds to EMR2 by using the proximity ligation assay (PLA). In FHR1 deficient patient samples no PLA signals were detected which underlined the specificity of the antibodies.

To verify the specificity of interaction we performed a new experiment and used a second specific antibody (TGF β R1) in order to demonstrate that FHR1 interacts with EMR2 but not with TGF β R1. To this end, monocytes and PBMCs were isolated from human blood of the same individual. Necrosis of PBMCs was induced by incubation at 65 °C for 45 min and subsequently these necrotic cells were incubated in normal human blood from the same donor. Using PLA assay the staining demonstrated that FHR1 interacts with EMR2 but not with TGF β R1 on monocytes. The control data are now shown in Fig. A below and in Supplementary Figure 3h in the manuscript.

Figure A FHR1 interacts with EMR2 but not with TGF β RI

Necrotic cells were incubated in normal human blood and cells subsequently stained for interaction of FHR1 with EMR2 or TGF β RI using the PLA assay. Laser scanning microscopy is shown on the left. Size bar=10 μ m. Statistical evaluation of the interaction is shown on the right (total picture fluorescence intensity of single cells EMR2 n=11, TGF β RI n=1)

2. Are the necrotic cells linked to monocytes by FHR1 internalised.

Response

In our manuscript we followed the scientific challenge to identify the receptor on monocytes which binds FHR1 and mediates inflammation. Using different methods including RNAseq and inhibition experiments we identified EMR2 as the binding receptor of FHR1 (Fig. 4, manuscript). This interaction was confirmed by ELISA and staining experiments. Immobilized FHR1 on a microtiter plate or physiologically bound to necrotic cells, interacts with the EMR2 receptor on monocytes and induces signaling cascades leading to NLRP3 induction. In addition, we now determined IL1 β concentrations when monocytes were incubated with immobilized FHR1 and phagocytosis was inhibited by using Latrunculin A. In this case bound FHR1 also induced IL-1 β release (Figure B below) Thus, internalization of FHR1 is not required for the induction of NLRP3.

Figure B Phagocytosis is not required for induction of IL-1 β in monocytes by FHR1

FHR1 (5 μ g/ml) was immobilized to a microtiter plate and incubated with blood monocytes in normal human serum (10%) with or without the phagocytosis inhibitor Latrunculin A (60 nM, Sigma-Aldrich). After 20h incubation, the IL-1 β concentration was measured in the supernatant. The data show mean \pm SEM of three independent experiments and donors (unpaired two-tailed t-test, $p \leq 0.05$).

3. It would be interesting to know whether FHR1 bound to necrotic cells or whether it also bound apoptotic ones, and if so whether the induces pro-inflammatory monocyte responses or abrogates the anti-inflammatory response they normally evoke.

We thank the reviewer for this question as we were interested in the same topic and have already performed binding experiments accordingly. We incubated human endothelial cells with staurosporine to induce apoptosis and secondary necrosis and determined binding of purified and natural FHR1 to the cells. Purified FHR1 and natural FHR1 from NHS bound to Annexin and Viability Dye positive secondary necrotic cells (An+/VD+), but not to early apoptotic cells (An+/VD-) or double negative cells (An-/VD-). This is in agreement with the fact that necrotic but not

apoptotic or living cells induce inflammation. The results are shown in Fig. C below and in manuscript Supplementary Fig. 3b.

Figure C recombinant and natural FHR1 bind to secondary necrotic, but not to early apoptotic or untreated cells

HUVEC cells (4.5×10^5) were incubated with staurosporine over night and incubated with recombinant FHR1 or NHS. Binodng of FHR1 was followed by flow cytometry. The data show mean values \pm SEM of three independent experiments (unpaired two-tailed t-test, ** $p \leq 0.01$, *** $p \leq 0.001$).

4. It is an over-interpretation to state that Figure 7a shows that “FHR1 attaches to necrotic cells surrounding fresh fibrinoid necrotic lesions in the glomerulus”. The present images merely show that FHR1 is deposited in necrotic lesions – in common with all other deposited complement proteins the mechanism of deposition/retention in the glomerulus cannot be ascertained from these light microscopy images. There are many other reasons why FHR1 could be deposited including by binding FH or C3b, dimerising with FHR2 etc, etc. Accordingly, it would be valuable to know which other complement components were deposited on the same location, including FH, FHR2 and C3. The absence of binding in FHR1 deficient individuals merely tells one that the antibody used was specific and informative about the biology of FHR1, although in concert with staining for other complement proteins it could be.

The expression ‘FHR1 attaches to necrotic cells surrounding fresh fibrinoid necrotic lesions in the glomerulus’ is now changed to ‘FHR1 is deposited in proximity to necrotic lesions’. To provide evidence that the deposition of FHR1 is also independent of other complement proteins and likely does not represent uncharacteristic trapping, we performed new staining experiments. In this case we stained tissue sections from FHR1 positive ANCA patients using FHR1, C3c and FHR2 antibodies. Besides a weak co-positivity of FHR1 and FHR2 with C3 in some areas, FHR1 was predominantly detected in close proximity to necrotic lesions (Fig. D below and manuscript Supplementary Fig. 4a).

Figure D FHR1, C3, and FHR1 staining in AAV kidney biopsies

Granular positivity for FHR1 (arrows, left panel) in parts of the glomerular matrix that are in proximity to fibrinoid necrosis („*“ = fibrin precipitates). In contrast, almost no positivity for C3c (middle panel) and FHR2 (right panel) is almost negative in these areas (same glomerulus). „>“ marks likely uncharacteristic serum positivity in the glomerular capillaries and „+“ marks (possibly uncharacteristic) positivity in the tubulointerstitium. Monoclonal FHR1 antibodies, C3c antiserum (Dako) and monoclonal FHR2 antibodies were used.

5. Figure 7c the authors use the PLA to identify EMR2/FHR1 complexes renal biopsies from patients with AAV. These clearly demonstrate a signal in FHR1 sufficient individuals but not in deficient ones (Δ FHR1/3). This is a good negative control but not one for specificity: this requires substituting the anti-EMR antibody for one specific for a different monocyte surface receptor.

The specificity of the interaction between FHR1 and EMR2 is demonstrated in new tissue stainings (AAV patients), showing the interaction of FHR1 with EMR2 but not with TGF β RI using PLA assay (Figure E below and Supplementary Figure 4c)

Figure E FHR1/EMR2 but no FHR1/TGF β RI complexes are detected in AAV kidney biopsies

EMR2/FHR1 complexes, but no TGF β RI/FHR1 complexes formed in kidney biopsies of AAV patients shown by PLA assay. DNA is stained with DAPI. Bar = 10 μ m.

6. Figures 7 d and e show that the IL-1beta and CRP concentrations are significantly higher in FHR1 sufficient individuals with AAV than Δ FHR1/3 ones but there is considerable overlap between the two groups that differ greatly in size. Accordingly, the results need to be interpreted with extreme caution, not least because of potential ascertainment biases caused by the small number of Δ FHR1/3 individuals studied and possible differences in the stage of the disease or concurrent treatment. More clinical data should be included to mitigate the possibility of ascertainment bias.

The reviewer is correct, the number of FHR1 deficient patients is rather low, however reflects the average percentage in a European cohort (Holmes et al. Determining the population frequency of *CFHR3/CFHR1* deletion at 1q32, 2013, Plos One 8/e60352). To address the concerns by Prof. Rees, we now determined the FHR1 concentration in each AAV serum sample and related the concentrations to different disease parameters. The analysis revealed a significant higher concentration of FHR1 in the patient samples compared to serum concentrations of FHR1 in healthy individuals (FHR1: 26.5 ± 2.3 μ g/mL, n=55 controls; FHR1: 69.8 ± 3.7 μ g/mL, n=313 patients; ***p<0.001) (Fig. F below and manuscript Fig. 7f). Furthermore, FHR1 concentrations negatively correlated with the glomerular filtration rate in the cohort (Figure G, below and manuscript Fig. 7g) and positively correlated with relapses (Fig. H below and now manuscript Fig.7h) and death rates (Fig. I and now manuscript Supplementary Fig. 4j). Also, the IL-1 β concentrations increased with FHR1 concentrations in patients without immune pretreatment (Fig. J and manuscript Supplementary Fig. 4i). The new data underline our findings that FHR1 concentrations are associated with inflammation and the progression of the disease.

Figure F, G, H, I, J FHR1 concentrations in serum samples of AAV patients correlate with disease progression

F: FHR1 concentrations in healthy individuals and AAV patients (unpaired two tailed t-test, Welch's correction).

G: Estimated glomerular filtration rate (eGFR) decreases with elevated FHR1 concentrations (Spearman correlation).

H, Percentage of relapses among AAV patients increases with elevated FHR1 concentrations.

I: The percentage of deaths of AAV patients increases with growing FHR1 levels.

J: Correlation between IL-1 β and FHR1 concentrations in AAV patients without immunosuppression therapy (Spearman correlation).

7. Figure f and g demonstrate the Δ FHR1/3 individuals have significantly lower proportion of crescents and segmental necrotic lesions. Again more clinical data and a validation cohort are required before definitive conclusions can be made about the possible direct effect of FHR1. Equally important, even a direct effect could be due to FHR1 functions other than its binding necrotic cells; for example, its reported direct inhibitory effects of C3 and C5 convertases (the latter especially pertinent because of the therapeutic efficacy of inhibition of the C5a receptor in nephritis in AAV). Similar comments apply to reported relapses in Δ FHR1/3 individuals.

We agree on the critical point mentioned by Prof. Rees that we have a low number of Δ FHR1/3 individuals in the cohort. However, AAV is a very rare disease (46-184 in 1×10^6 , Watts et al. Classification, epidemiology and clinical subgrouping of antineutrophil cytoplasmic antibody (ANCA)-associated vasculitis, 2015, Nephrol Dial Transplant, 30 Suppl 1:14-22) and the here described cohort is extraordinarily huge under these circumstances. As mentioned before we addressed the concerns and determined the FHR1 concentrations in each AAV patient and found a significant higher amount of FHR1 in the serum samples of the AAV patients compared to healthy control individuals (see also point 6 above). As shown in Figure F-J elevated FHR1 concentrations correlate with the progression of the disease. These data substantially extend the comparison between Δ FHR1/3 (n=11) and non-deficient patients (n=314) and demonstrate the involvement of FHR1 in AAV.

Although FHR1 is a complement protein, we here describe a new, complement independent function of FHR1 in inflammation by binding to necrotic type cells and in triggering an inflammatory response. Thereby we do not exclude the major contribution of complement in AAV and its role in activation of the C5-convertase as well as cleavage. C5a is another strong trigger of inflammation and recruits immune cells. We assume that complement activation leads to the activation of neutrophils that induces vascular injury with the subsequent binding of FHR1 to the damaged/necrotic cells in the injury and the amplification of inflammation. Thus, combined inhibition of FHR1 plus C5/C5a or C5aR1 is expected to be even more beneficial for the patient. The complement regulatory role of FHR1 (regulation of C5 but not C3) depends on the deposition of C3b and subsequent binding of FHR1 to C3b via the C-terminal domain. However, C3b bound FHR1 lacks inflammatory functions.

8. Of interest, the authors cite the repeatedly validated protective effect of in Δ FHR1/3 on susceptibility to another type of glomerulonephritis, mesangial IgA disease. Recent analyses of large cohorts have shown no detectable effect on severity of IgA nephritis, which contrasts with the authors 'suggestion of the opposite in AAV in which the major effect is on severity and/or progression. Data on as yet unreported effects of in Δ FHR1/3 on genetic susceptibility should be available in the public datasets from the European and North American GWASs published over the past.

In a recent paper by Medjeral-Thomas et al (Circulating complement factor H related proteins 1 and 5 correlate with disease activity in IgA nephropathy, 2017, Kidney Int, 92, 942-952) the authors reported that FHR1 plasma concentrations are elevated in an IgAN cohort and negatively correlated with the glomerular filtration rate. Similar results were reported in a study of Tortajada et al. (Elevated factor H related protein 1 and factor H pathogenic variants decrease complement regulation in IgA nephropathy, 2017, Kidney Int, 92, 953-963). The investigators measured elevated

FHR1 concentrations in IgAN and showed an association with disease progression. However, they hypothesized that FHR1 elevation deregulates factor H functions.

In summary, it is important to address the clinical issues raised. In some cases this will require new experiments or additional data but in many cases they could be remedied with by drawing more conclusions limited to the data. Perhaps changes to the introduction to set out the context succinctly might also help.

Andrew Rees

We thank Prof. Rees for the detailed critical reading of the manuscript, the fruitful suggestions and ideas and we hope to have answered and addressed all his questions.

Reviewer #2 (Remarks to the Author):

This paper by Irmischer et al describes a comprehensive sets of experiments to support the central hypothesis that FHR1, or factor H related protein 1, plays a role in sterile inflammation by binding to necrotic cells and triggering EMR2 activation and signaling, ultimately to inflammasome NLRP3 induction and cytokine production (IL-1-beta and TNF-alpha etc). The study is quite mechanistic and of considerable depth, employing a number of approaches and correlates, including dissection of signaling pathways, RNA seq analysis of gene regulation in FHR1-stimulated monocytes and human patient population analysis). The findings and conclusions are significance and could potentially explain why FHR1/3 gene deficiency is protective in AMD and IgAN. There are however a number of issues remain to be addressed and clarified.

1. The authors presented data showing serum is required for the FHR1 activity and the serum element is heat-sensitive, but there is no information or discussion about why serum is required and what are the essential serum element(s) and how it works. The authors have excluded complement being the required heat-sensitive element.

Response

In our manuscript we focused on the identification of the function of FHR1 and found a new, complement independent activity of FHR1 in inflammation via the binding to necrotic cells. We determined the activation of NLRP3 by FHR1 and identified the responsible G protein coupled receptor EMR2 on monocytes. However, a second signal from the serum is necessary to induce NLRP3 by FHR1. Using multiple inhibitors we were able to exclude several receptors but also complement activation as well as K^+ efflux. The identification of the serum component that likely co-activates the GPRC EMR2 is so far unclear. Biochemical and structural biology investigators have worked on the molecular basis of GPCR signaling, but how GPCRs translate

extracellular signals into biological responses is still elusive. It has been suggested that the high efficiency and specificity observed in GPCR signaling, depends on so called nanodomains on the plasma membrane or even form stable complexes with G proteins and effectors. (Calebiro, D., Koszegi, Z., *The subcellular dynamics of GPCR signaling*, 2019, *Molecular and Cellular Endocrinology* 483, 24-30). Others reported about proteolysis of EMR2 at a highly conserved cleavage site (Ser⁵¹⁸) occurring inside the endoplasmatic reticulum compartment resulting into two protein subunits that associate non-covalently as a heterodimer. (Lin et al. *Autocatalytic cleavage of the EMR2 receptor occurs at a conserved G protein-coupled receptor proteolytic site motif*, 2004, *Journal Biological Chemistry* 279, 31823-32). To clear the precise activation events of the receptor itself is out of the range of this manuscript. The topic of GPCR activation is now mentioned in the discussion part.

2. FH 18/19/20 domains are highly homologous to SCR3-5 of FHR1 and it was shown FH 19-20 can do the same thing as FHR1. Can whole FH do the same thing as FHR1, i.e binding to necrotic cells and triggering cytokine production in monocytes?

Response

The reviewer is correct, the domains of Factor H SCRs 18/19/20 are highly homologues to SCRs 3/4/5 of FHR1. When FH SCR19/20 were immobilized to a microtiter plate and incubated with monocytes in NHS, IL-1 β was induced in monocytes (manuscript Fig. 1f). However, immobilized full length Factor H did not induce IL-1 β (manuscript Fig. 1b), indicating that the C-terminus is inhibited by for example back folding, as mentioned in the discussion part. Furthermore, Factor H in contrast to FHR1 does not directly bind to necrotic cells (manuscript, Fig. 5a & Supplementary Fig. 3a) and does not induce IL-1 β (manuscript, Figure 5i).

3. While the data and hypothesis presented could potentially explain why FHR1/FHR3 gene deletion is protective for AMD, IgAN and ANCA, the opposite association, i.e. FHR1/FHR3 deficiency increases the risk of SLE, aHUS and RA cannot be explained. What's are the author's thoughts on the latter?

Presenting a new function of FHR1 raises the question how this function fits to the genetic deletion of *CFHR3/CFHR1* genes regarding the risk for specific diseases. In AMD and IgA the deletion of *CFHR3/CFHR1* is protective and in SLE, aHUS and RA the same deletion increases the risk. The deletion in aHUS is combined with the presence of autoantibodies to FH. Our ideas are, that FHR1 has at least two functions: one by binding to C3b and regulating complement, and the other, which is independent of complement, by binding to necrotic cells and inducing inflammation. These two functions may determine the detrimental effects of a *CFHR1* gene deficiency in diseases.

4. page 19, line 366-367, why FHR1 antibodies would augment rather than reduce the inflammatory response in NHS? Similarly on the same page, line 369-370, "In conclusion, inhibition of the FHR1 N-terminus augmented FHR1 induced inflammation...." seems to contract the sentence immediately before it which stated " Monoclonal FHR1 F(ab)₂reduced the IL-1 β release....by about 50%.." This paragraph is very confusing
Wording of this paragraph is corrected.

Reviewer #3 (Remarks to the Author):

In their paper, Irmischer, Brix and Zipfel et al. present data on necrotic cell-induced inflammation. They demonstrate that FHR1 specifically induces NLRP3-mediated IL-1 β maturation and secretion when immobilized. They further suggest that FHR1 binds to EMR2 (ADGRE2), mutations in which are associated with bacteria-induced urinary tract infections, the defense against which obviously involves necrotic type cell death. They use IF to detect the FHR1/EMR2 complex on the surface of what they refer to as necrotic cells. They confirm a previously described MDA-LDL binding of FHR1 (published in Weismann et al., nature 2011, ref 17). Finally, they study FHR1 in samples from AAV and AS patients. Unfortunately, the study is entirely correlative and does not answer the major questions raised by the correlative data, e.g. how FHR1 drives inflammasome assembly or how necrotic cells in tissues might specifically bind to this factor. The role of EMR2 is touched upon in a very superficial manner.

Methodologically, several severe issues must be raised (see below), such as the induction of necrosis, the methods to evaluate the inflammasomes and the detection of necrosis in tissue samples including the lack of validation of antibodies in knockout mice. In addition, the authors ignore the readily available mouse models of AAV which would be very useful for mechanistic insights in this study.

Importantly, the cell culture models of necrosis-induction are highly artificial (heat stress only). To be more convincing, a method of regulated necrosis, e.g. pyroptosis, should have been chosen for the investigation of NLRP3-mediated necrosis (as obviously present in some other figures).

In summary, this is an interesting but entirely correlative study, performed with limited expertise to the field and poor methodology. I cannot recommend publication in Nature Communications.

Response

We thank this reviewer for the comprehensive analysis of our manuscript and the constructive suggestions. We address all concerns by this reviewer in the following points

In our manuscript we identified a new function of a member of a family of complement proteins, which is associated with a number of diseases. We showed by multiple different assays that this protein binds to necrotic cells via oxidized lipid *in vitro* and *vivo* and that FHR1 as well as the murine homologue FHRB in its bound form induces pro-inflammation in monocytes. We subsequently identified the binding receptor (EMR2) of FHR1 and showed NLRP3 activation, caspase 1 activation as well as upregulation and release of IL-1 β , by multiple assays.

The induction of NLRP3 and high IL-1 β expression were reported in regulated cell death pathways such as pyroptosis or necroptosis (Bergsbaken et al., *Pyroptosis: host cell death and inflammation*, 2009, *Nature reviews microbiology* 7, 99-109). We were aware of the multiple pathways of NLR stimulation in monocytes and did not neglect pyroptosis or necroptosis and the action of GSDMD. Because of these described mechanisms we determined cell death rates upon incubation of the monocytes by measuring the LDH release of the treated monocytes. After prolonged incubation with FHR1 (20 h) cells stayed intact and no LDH increase was measured in the supernatant (manuscript Fig. 1g). In addition, ROS was not induced by FHR1 (manuscript Supplementary Figure 2k) and IL1 β was released over time and not suddenly by cell disruption (manuscript Figure 1e). Therefore, cell death of monocytes such as pyroptosis and necroptosis was excluded after incubation with FHR1 for more than 20 h.

As this reviewer mentioned repeatedly pyroptosis or necroptosis we addressed the concerns of this reviewer and further evaluated cell death in response to FHR1. New assays were performed where we determined cell viability using the cell titer blue assay. Incubating the cells with immobilized FHR1 for 20 h revealed viable cells, in contrast to nigericin treated cells, a trigger of pyroptosis (Figure K below and manuscript Fig. 1g). In addition, we incubated FHR1 with monocytes with or without the necroptosis inhibitor Necrostatin-5 and observed no significant change regarding the IL-1 β release (Figure L below). Thus, FHR1 induces IL-1 β release, but does not induce cell death in monocytes. Activation of NLRP3 in monocytes/macrophages does not ultimately result in cell death. Recently Evavold et al. (*The pore-forming protein gasdermin D regulates Interleukin-1 secretion from living macrophages*, 2018, *Immunity* 48, 1-10) reported a non-pyroptotic function for gasdermin D in response to host-derived oxidized lipids. In this case gasdermin D forms pores for the secretion of cytosolic cytokines under conditions of cell hyperactivation without cell death.

Figure K-L FHR1 does not induce cell death

K: Monocytes remain healthy after incubation with FHR1, as demonstrated by full cell viability, measured via cell titer blue assay (unpaired-two-tailed t-test, *** $p \leq 0.001$).

L: FHR1 induces IL-1 β in monocytes in presence of the necroptosis inhibitor Necrostatin-5 (100 μ M, abcam) (unpaired-two-tailed t-test).

Major concerns

- How does FHR1 drive inflammasome activation? This requires the analysis of inflammasome components (ASC, NLRP3, caspase-1 cleavage), but also detection of downstream markers and necrosis (pyroptosis) mediated by GSDMD.

Using RNAseq and a number of inhibitors we identified the activation pathway of FHR1 in monocytes. We found that NLRP3 was induced via the G protein-coupled receptor EMR2, which was confirmed by binding and inhibition assays (manuscript Fig. 4d-h). EMR2 induces the PLC pathway with subsequent increase of intracellular calcium, which activates the NLRP3 inflammasome (Fig. 3f, g). Cell death of monocytes such as pyroptosis or necroptosis was excluded by LDH measurement and now also by cell viability as well as inhibition assays (see manuscript Fig. 1g and Figure K and L above). In addition our RNAseq data showed even a significantly downregulation of GSDMD in FHR1 treated monocytes compared to BSA treated monocytes (not included in the manuscript).

- How does EMR2 regulate pyroptosis signalling? The EMR2 receptor undergoes autocatalytic cleavage, producing an extracellular subunit that noncovalently binds a transmembrane subunit. Is this signalling involved in inflammasome activation, ASC-Spec formation, caspase-1/11 activity and GSDMD-cleavage? Is GSDMD required for IL-1 β release? A simple crKO of EMR2 would help to link or unlink the underlying mechanisms here. However, a cleavage deficient mutant knock-in would be required for mechanistic links to be concluded.

FHR1 does not induce pyroptosis or necroptosis in monocytes (see manuscript Fig. 1g and Figure K and L above).

- There are mouse models available and widely used for the study of AAV. How would a EMR2-ko/cleavage-deficient mutant and/or an FHR1 (if present in mice)-ko respond to a model like this?

There is a mouse model for AAV, that is correct, however, there is no mouse model for EMR2 (mice do not express EMR2) and so far there does not exist a FHRB ko mouse model. Therefore, we could not follow mouse experiments as suggested by the reviewer. Instead FHR1 functions were characterized in a large AAV cohort.

- Interestingly, the authors, as much as they represent experts in the field of complement, appear to ignore the literature on necrosis and inflammasomes. They ignore the obvious connection to gasdermin D/pyroptosis despite several high impact papers were published in the recent 3 years on this topic. They also do not appear to be aware of the novel concepts on necroinflammation which would be critical if the title remains unchanged.

As demonstrated by additional experiments and explained in detail in the previous paragraphs, monocytes do not undergo a form of cell death (pyroptosis or necroptosis) upon incubation with immobilized FHR1. Thus FHR1 does not induce so called necroinflammation (Linkermann A, Death and fire – the concept of necroinflammation , 2018, Cell Death & Differentiation 26, 1-3).. However, as shown in the manuscript in detail, FHR1 binds to necrotic type cells and acts as a central mediator of inflammation. This is now mentioned in the introduction and discussion.

- Induction of necrosis is performed by simply heating the cells to 63°C. This is a highly artificial model. Both AAV and AS have been demonstrated to involve processes of regulated necrosis (pyroptosis, necroptosis, parthanatos and others). Why do the authors work on artificial necrosis instead of inducing pyroptosis or necroptosis in their systems. These pathways may be associated with a different immune response/DAMP release compared to simply heat stress. Especially when putting IL-1b on the central platform in Fig. 1 and 2, why not work on pyroptosis induced by e.g. nigericin?

Necrotic cells are shown to bind via oxidized lipids the human plasma protein FHR1 that subsequently induces inflammation via monocytes. As suggested by this reviewer we specifically induced apoptosis/sec. necrosis, necroptosis and pyroptosis and measured binding of FHR1 to these cell surfaces. FHR1 bound to pyroptotic, necroptotic and secondary necrotic cells but not to living or early apoptotic cells (Figure M-O below and manuscript Supplementary Figure 3b-3d).

Figure M-O Binding of FHR1 to cells with different type of cell death

M: Recombinant FHR1 as well as FHR1 from NHS bind to Staurosporine induced secondary necrotic HUVEC cells, which are Annexin (An) and Viability Dye (VD) positive (An+/VD+), but not to early apoptotic (An+/VD-) and double negative (An-/VD-) cells, as shown by flow cytometry using a monoclonal FHR1 antibody.

N: FHR1 binds to necroptotic U937 cells (induced by Z-VAD-FMK, TNF α and Cycloheximide), which are Annexin positive (An+), but not to Annexin negative cells (An-), as shown by flow cytometry using the monoclonal C02 antibody.

O: FHR1 binds to Nigericin induced pyroptotic THP1 cells (An+/VD+), but not to double negative (An-/VD-) cells, as shown by flow cytometry using the monoclonal C02 antibody.

- How do the authors explain the upregulation of TNF α and IL-6 in Fig. 1j and 1k? Are these simply elements of a cytokine storm or is this a highly specific upstream signal from necroptotically dying cells? If the latter is the case, necroptosis should be studied in the presence and absence of FHR1.

TNF α and IL-6 are upregulated in parallel to IL-1 β . Upregulation of TNF α and IL-1 β is in a time dependent manner (manuscript Figs 1e and 1m), and increase with immobilized FHR1 concentrations (manuscript Figs 1d). Necroptosis is excluded (Figure L above).

Minor remarks

- The discussion and introduction should include the recent literature/concepts on pyroptosis (GSDM-mediated necrosis) and its influences on the immune system.

The information about gasdermin-mediated necrosis is now added to the introduction and discussion parts.

- The resolution of PLA is insufficient. This is a complex technique that should be used in addition to others only if very good primary antibodies are available which does not seem to be the case here.

The resolution of the PLA in the original picture is very high, which probably became lost in this combined version of submitted manuscript. The quality will be much higher, once the Figures are submitted separately. The used primary antibodies are of best quality available.

• Fig. S4 – Fig. S4A does not allow any conclusion in the presented quality. How representative is the western blot in Fig. S4G?

Fig. S4A (now Fig. S4B) is in a much better quality than presented in the version of submitted manuscript. Western blot in S4G (now Fig. S4M) is representative of at least 3 independent experiments. We added this information to the legend.

Reviewer #4 (Remarks to the Author):

Irmischer S et al identify a new receptor for FHR1. Moreover, they observed that FHR1 binds to necrotic cells to exacerbate pro-inflammatory cytokine production/release in monocytes. This work is important and describes a mechanism associated with persistent inflammation in disease like atherosclerosis and AAV, where the accumulation of necrotic cells is a critical pathological factor.

I think this body of work is thorough and I truly enjoyed learning from this manuscript. I only have minor comments.

We also thank this reviewer for the detailed analysis of the manuscript and are happy to answer all questions

Comments:

1) Excellent aspects about the work:

--This work focuses on human cells and human tissues therefore the translational relevance is very clear.

--The authors use RNA seq and functional in vitro assays to help determine a new receptor for FHR1 on monocytes.

2) Please revise the statistics throughout. In several cases two-tailed t test is not the best or most accurate statistical method for your data.

According to the reviewers comment we performed additional statistical tests to compare the CRP as well as IL-1 β levels in AAV patients with or without FHR1 and healthy individuals (Figure 7d, e). We performed the Kruskal-Wallis Test which revealed for CRP a p-value of $\leq 0,001$ (***) , which is comparable to the p-value $\leq 0,001$ (***) between patients with and without FHR1 calculated by the unpaired two-tailed t –test with Welch's correction. Similarly, the p-values for IL-1 β induction calculated with these two tests did not differ. The Kruskal-Wallis Test calculated a p-value =0,0251 (**) and the unpaired two-tailed t –test with Welch's correction a p-

value =0,0188 (**) between patients with and without FHR1. The new tests are now integrated into the manuscript.

3) Given that apoptotic cells are also present in atherosclerosis and AAV do you know if FHR1 binds to apoptotic cells? Apoptotic cells do not evoke inflammation and exert a tissue reparative response on monocytes/macrophages. If FHR1 does not bind to ACs, then your new mechanism may also describe why necrotic cells evoke pro-inflammation on monocytes and why ACs do not.

We were interested in the same question and performed binding assays with FHR1 binding to apoptotic and secondary necrotic cells. Both recombinant and natural FHR1 bound to secondary necrotic cells but not early apoptotic cells. The results are shown in Figure P below and in manuscript Supplementary Fig. 3b)

Figure P recombinant and natural FHR1 bind to secondary necrotic, but not to early apoptotic cells

HUVEC cells were incubated with staurosporine over night and incubated with recombinant FHR1 or NHS. Staining of the cells was performed with Annexin (An) and Viability Dye (VD) to follow the stages of early apoptosis (An+/VD-) and secondary necrosis (An+/VD+). The data show mean values \pm SEM of three independent flow cytometry experiments (unpaired two-tailed t-test, ** $p \leq 0.01$, *** $p \leq 0.001$).

4) the necrotic cells were generated via heat-stress. Do necroptotic cells also bind FHR1? Do pyroptotic cells bind FHR1?

FHR1 bind to necroptotic and pyroptotic cells. The results are shown in Figures Q and R below and in manuscript Supplementary Fig. 3c, d).

Figures Q-R FHR1 binds to necroptotic and pyroptotic cells

Q: FHR1 binds to necroptotic U937 cells (induced by Z-VAD-FMK, TNF α and Cycloheximide), which are Annexin positive (An+), but not to Annexin negative cells (An-), as shown by flow cytometry using the monoclonal C02 antibody (unpaired two-tailed t-test, ** $p \leq 0.01$).

R: FHR1 binds to Nigericin induced pyroptotic THP1 cells (An+/VD+), but not to double negative (An-/VD-) cells, as shown by flow cytometry using the monoclonal C02 antibody (unpaired two-tailed t-test, ** $p \leq 0.01$).

5) Do FHR1 knockout mice exist? Doesn't appear so, based on a cursory search. If they become available in the future, then this work will lay the foundation for causative experiments in which one can test whether the lack of FHR1 can diminish necrotic cores in atherosclerosis and any disease where necrosis underpins the pathology.

Today no FHRB knockout mice exist.

Reviewers' comments:

Reviewer #1 (Remarks to the Author):

The authors have been very responsive to all the referees critiques, both with new experimental data and clarifications; indeed it is a model of how revisions should be done! Specifically the authors have considered carefully and addressed convincingly the issues and uncertainties I raised. Their new data all all supportive of their original conclusions and it is appropriate that most should be confined to the supplemental figures.

In short, the authors' data identifies a novel biological function for the FHR1 component of complement and provide convincing evidence for its involvement in vivo, including in human inflammatory disease - ANCA-associated vasculitis but with implications well beyond this specific context. It is an important contribution for which the authors should be congratulated.

Reviewer #2 (Remarks to the Author):

The authors have adequately addressed the issues I raised and and I have no further comments

Reviewer #3 (Remarks to the Author):

In the revised version of the manuscript, still many of the concerns raised remain unanswered. In particular:

1) There is no experiment on "how FHR1 drives inflammasomes assembly". The authors state that they identified EMR2 as the binding receptor, but no binding of FHR1 to EMR2 is demonstrated – but it could be by a simple IP experiment. Instead, the authors claim to use a specific inhibitor of EMR2, but this is not convincing. Where do the authors demonstrate the specificity of the inhibitor? Why can't they knock out or knock down the receptor and demonstrate the absence of the FHR1-induced calcium flux?

2) There is no evidence on "how necrotic cells in tissues might specifically bind to this factor". The authors have provided other ways of driving cells into necrosis (e.g. secondary necrotic after intrinsic apoptosis and TZ for necroptosis) which is highly appreciated and promising. However, the authors do not answer "how necrotic cells might specifically bind to this factor". What is it in the necrotic cells that is sensed? This question remains unaddressed.

3) The methods of necrosis induction were extended on nigericin. It is understood that EMR2 does not induce cell death, but it does result in inflammasomes signaling. In this sense, a GSDMD-ko should be provided to demonstrate if IL-1b release depends on GSDMD.

a. Minor: The detection of dead cells by "cell titer blue assay" is by no means an established technique.

b. Minor: The idea to prevent IL-1b release by the addition of a necrostatins makes no sense. Of course, RIPK1 (the target of Nec-5) is not the critical regulator of IL-1b release!

c. Minor: "cell viability" is an interpretation (new Fig. 1G). Please label the axis with what you measured – cell titer blue positivity?

4) Antibodies have not been validated, as requested.

5) Mouse models. It is understood that mice do not express EMR2. However, this makes it even more important to provide human knockout cell lines of EMR2 to validate the antibody used. The cells in which the specificity of the EMR2 inhibitor have been tested should provide an ideal example.

Reviewer #4 (Remarks to the Author):

The authors addressed my concerns and I think the work is suitable for publication.

Reviewers' comments:

Reviewer #1 (Remarks to the Author):

The authors have been very responsive to all the referees critiques, both with new experimental data and clarifications; indeed it is a model of how revisions should be done! Specifically the authors have considered carefully and addressed convincingly the issues and uncertainties I raised. Their new data all all supportive of their original conclusions and it is appropriate that most should be confined to the supplemental figures.

In short, the authors' data identifies a novel biological function for the FHR1 component of complement and provide convincing evidence for its involvement in vivo, including in human inflammatory disease - ANCA-associated vasculitis but with implications well beyond this specific context. It is an important contribution for which the authors should be congratulated.

Reviewer #2 (Remarks to the Author):

The authors have adequately addressed the issues I raised and and I have no further comments

Reviewer #3 (Remarks to the Author):

In the revised version of the manuscript, still many of the concerns raised remain unanswered.

In particular:

1) There is no experiment on;how FHR1 drives inflammasomes assembly;. The authors state that they identified EMR2 as the binding receptor, but no binding of FHR1 to EMR2 is demonstrated; but it could be by a simple IP experiment. Instead, the authors claim to use a specific inhibitor of EMR2, but this is not convincing. Where do the authors demonstrate the specificity of the inhibitor? Why cant they knock out or knock down the receptor and demonstrate the absence of the FHR1-induced calcium flux?

Response

In our manuscript we identified the complement protein FHR1 as an inducer of inflammation in monocytes.

EMR2 involvement in this inflammatory pathway was identified by RNAseq analysis of monocytes. Subsequently, interaction of FHR1 with EMR2 was confirmed by several experiments and methods:

- Binding of recombinant FHR1 to recombinant EMR2 (ELISA)(Fig. 4g)
- antibody independent real time interaction of FHR1 with recombinant EMR2 (Blitz)(Fig. 4h)
- Inhibition of induction of monocytes (IL-1 β) by FHR1 using anti EMR2 (activation assay)(Fig. 4f)

- Binding of natural FHR1 to EMR2 on blood monocytes (complex formation shown by proximity ligation assay) interaction was not seen in FHR1 deficient serum and specificity of the EMR2 antibody is demonstrated by lacking signals when anti EMR2 was replaced by anti TGFbR1 (Fig. 5j, Suppl. Fig. 3h).
- Complex formation (again by PLA assay) between EMR2 and FHR1 was confirmed in tissue sections from ANCA and atherosclerosis patients. FHR1 deficient tissue sections showed no signals. (Fig. 7C, Suppl. Fig. 4b-c)

Interaction of FHR1 with EMR2 needs immobilization of FHR1. Plasma circulating full length FHR1 does not bind to EMR2 on monocytes. Therefore, an immunoprecipitation in fluid phase is not feasible. The FHR1 molecule needs to bind to the necrotic cell and then binds to the EMR2 receptor on monocytes. This is exactly shown by the proximity ligation assay in tissue sections or on necrotic cells (Fig. 5j, Suppl. Fig. 3h, Fig. 7C, Suppl. Fig. 4b-c).

Proximity ligation assay (in situ PLA) is a technology to show protein interactions and modifications with high specificity and sensitivity. It is a relatively new method used to detect and localize protein targets in unmodified cells and tissues and thus can replace conventional co-immunoprecipitation and co-localization techniques. (Söderberg, et al. *Direct observation of individual endogenous protein complexes in situ by proximity ligation* (2006). *Nature Methods* 3, 995-1000, M. Jarvius, J. et al. : *In situ detection of phosphorylated platelet-derived growth factor receptor beta using a generalized proximity ligation method*. In: *Molecular & cellular proteomics* (2007): MCP. 6, Number 9, ISSN 1535-9476, S. 1500–1509). First, two primary antibodies raised in different species recognize the target antigen on the proteins of interest (here FHR1 and EMR2). Special secondary antibodies directed against the constant regions of the different primary antibodies, called PLA probes, bind to the primary antibodies. Each of the PLA probes has a unique short DNA strand attached to it. If the PLA probes are in close proximity (part of a protein complex), the DNA strands will be amplified by rolling circle DNA synthesis. Fluorescent-labeled complementary oligonucleotide probes bind to the amplified DNA. The resulting high concentration of fluorescence is easily visible as a distinct bright spot when viewed with a fluorescence microscope. Depending on the length of the oligonucleotides on the proximity probes, the distances vary between 10 to 30 nm, including the size of the two antibodies and the connecting oligonucleotides. The distance is comparable to the distance at which resonance energy transfer occurs between fluorophores (5 -10 nm). In situ PLA confers dual-binder specificity for protein detection in situ and can reveal interactions between proteins directly in normal cells and tissues without being subject to artifacts of overexpression or ectopic expression. Because samples are fixed before analysis, snapshots of cellular processes are obtained, and transient interactions of endogenous proteins can therefore be easily detected (Calippe et al. *Complement Factor H Inhibits CD47-Mediated Resolution of Inflammation*. *Immunity*. 2017;46(2):261-272).

FHR1 binds via its N-terminus to necrotic cells (Fig. 5f) and via its C-terminus to EMR2 on monocytes (Fig. A below). This assay confirms binding between FHR1 and EMR2 with different proteins and antibodies (in this case monoclonal cMyc antibody).

Fig. A: The C-terminus of FHR1 (SCR3-5) binds to EMR2

EMR2, BSA or gelatin (each 5 µg/mL) were immobilized, blocked and incubated with the recombinant FHR1 fragments SCR1-2 and SCR3-5 and stained with monoclonal cMyc antibody (R&D Systems, LOT: CVQ0617121). The C-terminus SCR3-5, but not the N-terminus SCR 1-2 of FHR1 bound to EMR2. n=3, unpaired two-tailed t-test, * p<0.05.

Specificity of the EMR2 antibody (R&D Systems, CAT AF4894) used in inhibition assays (Fig. 4f) was previously shown by Western blot analysis and inhibition studies. Antibody staining of THP1 lysate with EMR2 revealed only one band of the expected 90kDa (Fig. B). Furthermore, functional activity of the antibody was demonstrated by inhibition of cell adhesion (Fig. C).

REDACTED

REDACTED

Fig. D: FHR1 induced inflammation is independent of CD97

The inhibition of CD97, a receptor with high sequence homology of 97.5% to EMR2, had no effect on FHR1 induced IL-1 β release of monocytes. FHR1, BSA (each 5 μ g/mL) or LPS (5 ng/mL)

were immobilized and incubated with monocytes (1×10^5) for 20h at 37°C and 5%CO₂. n=3, different donors, unpaired two-tailed t-test, ** p≤0.01.

As suggested by this reviewer, we previously also tried to knock down the EMR2 receptor in blood derived monocytes by siRNA, but found out that these cells were no longer inducible for IL-1β secretion after the transfection procedure.

Fig. E: Transfected blood monocytes lose their ability to react on stimulation.

Primary monocytes (3×10^6) were transfected with either EMR2 siRNA (450 nM, Santa Cruz Biotechnology, sc-45381) or control siRNA-A (450 nM, Santa Cruz Biotechnology, sc-37007) using the Amaxa™ Human Monocyte Nucleofactor™ Kit (Lonza, VPA-1007). Subsequent treatment with 20 ng/mL LPS revealed no IL-1b signals). Bars represent 3 independent experiments with different donors.

In addition, we tested cell lines to knock down the EMR2 receptor, but, in contrast to freshly isolated primary monocytes, THP-1 cells behaved differently. As previously described, THP1 cells need two initial signals to induce IL-1β secretion, whereas primary monocytes react on a single stimulus (Elliott, EI. *et al. Immunity* 2016, 44(4): 713-5; Gaidt, MM. *et al. Immunity* 2016, 44(4): 833-46; Netea, MG. *et al. Blood* 2009, 113(10): 2324-35). Therefore, experiments with EMR2 knocked down primary cells or knocked down THP-1 cells regarding FHR1 signaling were not feasible.

Fig. F: FHR1 does not induce IL-1β in THP-1 cell

FHR1 or BSA (each 5 μg/mL) were immobilized and incubated with cells (1×10^5) for 20 h at 37°C and 5% CO₂ and IL-1β was measured in the supernatant. THP1 cells were treated with PMA (0.4 μg/mL). Single experiment.

2) There is no evidence on; how necrotic cells in tissues might specifically bind to this factor;. The authors have provided other ways of driving cells into necrosis (e.g. secondary necrotic after intrinsic apoptosis and TZ for necroptosis) which is highly appreciated and promising. However, the authors do not answer how necrotic cells might specifically bind to this factor;. What is it in the necrotic cells that is sensed? This question remains unaddressed.

Response

In Figure 6 of the manuscript we demonstrate binding of FHR1 to MDA-LDL (ELISA and real time interaction). Oxidization of lipids is typical on necrotic cells (Chang, MK *et al.* Proc Natl Acad Sci U S A 1999; 96(11): 6353-6358). We also showed that the N-terminus but not the C-terminus of FHR1 binds to MDA-LDL (Fig. 6e). Furthermore, natural FHR1 binds to MDA on necrotic cells (PLA assay Fig. 6d). The specificity of this interaction is demonstrated by lack of interaction signals on necrotic cells after they were incubated in FHR1 deficient serum. Thus, we clearly demonstrated a ligand for FHR1 binding on necrotic cells.

3) The methods of necrosis induction were extended on nigericin. It is understood that EMR2 does not induce cell death, but it does result in inflammasomes signaling. In this sense, a GSDMD-ko should be provided to demonstrate if IL-1b release depends on GSDMD.

Response

As suggested by this reviewer we induced cell death using nigericin to show binding of FHR1 also to these dead cells (Suppl. Fig. 3d). FHR1 binding to EMR2 induces inflammation but not cell death, leading to the release of IL-1β. As previously described by Evavold *et al.* (Immunity 2018; 48(1): 35-44.e6) IL-1β is either released by hyperactivated cells via gasdermin pores or by burst and death of the cell. As we excluded cell death and found the involvement of Caspase-1, which cleaves GSDMD, we concluded that IL-1β is released by GSDMD pores. Furthermore, involvement of GSDMD is already shown by caspase 1 inhibition in the manuscript, as all

GSDMD inhibitors we found inhibit Caspase-1. To work with knockout cell lines is again not feasible, as FHR1 does induce inflammation in freshly isolated primary cells, but not in THP-1 cells (see Fig. D above).

a. Minor: The detection of dead cells by cell titer blue assay; is by no means an established technique.

Response

There are several publications using the cell titer blue assay to show cell viability e.g.

- Nakagawa T. *et al.* (Nature 2005;434(7033):652-8),
- Jacobsen KM *et al.* (Cell Chem Biol. 2018; 25(11): 1337-1349.e12),
- Willems, T. *et al.* (J Virol Methods 2011; 173(1): 108-14),
- Quabazard, B. *et al.* (PLoS One 2013; 8(11): e80135),
- Idrees, A. *et al.* (Int J Artif Organs 2018; 41(11): 779-788).

b. Minor: The idea to prevent IL-1b release by the addition of a necrostatins makes no sense. Of course, RIPK1 (the target of Nec-5) is not the critical regulator of IL-1b release!

Response

To study necroptosis regarding FHR1 functions was suggested by this reviewer, therefore we performed the experiment with necrostatin, a common inhibitor of necroptosis.

c. Minor: cell viability; is an interpretation (new Fig. 1G). Please label the axis with what you measured; cell titer blue positivity?

Response

The y-axis is now labeled with “**CTB assay (% cell viability)**” according to the previous named publications using CTB before (see above).

4) Antibodies have not been validated, as requested.

Response

To assess specific binding between FHR1 and EMR2 we performed proximity ligation assays in whole blood and in tissue sections of AAV patients using the FHR1 antibody together with EMR2 and also an antibody specific for another surface receptor on monocytes (TGFβR1). In PLA control experiments with TGFβR1 and also in experiments with FHR1 deficient samples no signals were detected, demonstrating the antibody specificity for the detection of FHR1/EMR2 complexes (Fig. 5j, Fig. 7c, Suppl. Fig. 3h, Suppl. Fig. 4c).

5) Mouse models. It is understood that mice do not express EMR2. However, this makes it even more important to provide human knockout cell lines of EMR2 to validate the antibody used. The cells in which the specificity of the EMR2 inhibitor have been tested should provide an ideal example.

Response

To knock down the EMR2 receptor by siRNA is not feasible at the moment, as THP-1 cells are not a suitable cell line to work with for these experiments and freshly isolated primary monocytes lose their ability to release IL-1 β after the transfection procedure (see Fig. D-F above). An EMR2 knock out with CRISPR-Cas9 is excluded for the same reasons, as the cell line behaves differently and primary monocytes do not proliferate in culture, which would be necessary to expand the clone from a single cell.

Reviewer #4 (Remarks to the Author):

The authors addressed my concerns and I think the work is suitable for publication

REVIEWERS' COMMENTS:

Reviewer #3 (Remarks to the Author):

No further comments.